# Integrating analog and digital modes of gene expression at *Arabidopsis FLC*

Rea L Antoniou-Kourounioti[1,2†], Anis Meschichi[3†], Svenja Reeck[4†], Scott Berry[5], Govind Menon[1], Yusheng Zhao[6], John Fozard[1], Terri Holmes[7], Lihua Zhao[3], Huamei Wang[8], Matthew Hartley[9], Caroline Dean[4*], Stefanie Rosa[3*], Martin Howard[1*]

[1]Department of Computational and Systems Biology, John Innes Centre, Norwich, United Kingdom; [2]School of Molecular Biosciences, College of Medical, Veterinary and Life Sciences, University of Glasgow, Glasgow, United Kingdom; [3]Swedish University of Agricultural Sciences, Plant Biology Department, Uppsala, Sweden; [4]Department of Cell and Developmental Biology, John Innes Centre, Norwich, United Kingdom; [5]EMBL Australia Node in Single Molecule Science, School of Medical Sciences, University of New South Wales, Sydney, Australia; [6]State Key Laboratory of Plant Cell and Chromosome Engineering, Institute of Genetics and Developmental Biology, Chinese Academy of Sciences, Beijing, China; [7]Faculty of Medicine and Health Sciences, Norwich Medical School, University of East Anglia, Norwich, United Kingdom; [8]College of Life Sciences, Wuhan University, Wuhan, China; [9]European Molecular Biology Laboratory, European Bioinformatics Institute (EMBL-EBI), Wellcome Genome Campus, Cambridge, United Kingdom

*For correspondence:
caroline.dean@jic.ac.uk (CD);
stefanie.rosa@slu.se (SR);
martin.howard@jic.ac.uk (MH)

†These authors contributed equally to this work

**Competing interest:** The authors declare that no competing interests exist.

**Abstract** Quantitative gene regulation at the cell population level can be achieved by two fundamentally different modes of regulation at individual gene copies. A 'digital' mode involves binary ON/OFF expression states, with population-level variation arising from the proportion of gene copies in each state, while an 'analog' mode involves graded expression levels at each gene copy. At the *Arabidopsis* floral repressor *FLOWERING LOCUS C (FLC)*, 'digital' Polycomb silencing is known to facilitate quantitative epigenetic memory in response to cold. However, whether *FLC* regulation before cold involves analog or digital modes is unknown. Using quantitative fluorescent imaging of *FLC* mRNA and protein, together with mathematical modeling, we find that *FLC* expression before cold is regulated by both analog and digital modes. We observe a temporal separation between the two modes, with analog preceding digital. The analog mode can maintain intermediate expression levels at individual *FLC* gene copies, before subsequent digital silencing, consistent with the copies switching OFF stochastically and heritably without cold. This switch leads to a slow reduction in *FLC* expression at the cell population level. These data present a new paradigm for gradual repression, elucidating how analog transcriptional and digital epigenetic memory pathways can be integrated.

## Editor's evaluation

Regulation of gene expression in many biological systems occurs either digitally where gene expression is either on or off or through an analog mode with graded modulation of gene expression. In this study, the authors report how these two regulatory modes are integrated into a one-way switch pattern to control the expression of the *Arabidopsis* floral repressor gene *FLOWERING LOCUS C* (*FLC*). The results of their work lead the authors to propose that analog regulation in the autonomous flowering pathway precedes digital regulation conferred by Polycomb silencing before cold

exposure, and that this temporal switch correlates with the strength of transcription at the *FLC* locus in different genetic backgrounds.

## Introduction

One of the most fundamental questions in molecular biology is how quantitative gene expression is achieved. Traditionally, such regulation is ascribed to sequence-specific transcription factors that bind to regulatory DNA elements. According to the concentration of the transcription factors, gene expression can then be quantitatively up- or downregulated. While such regulation undoubtedly occurs in many systems, it has become abundantly clear in recent years that this paradigm is fundamentally incomplete. This is especially so in eukaryotes where quantitative transcriptional regulation can arise from modulation of the local chromatin environment of a gene. For example, by varying the type and level of histone modifications, DNA accessibility can be radically altered (*Ahmad et al., 2022*). In one scenario, nucleosome positioning affects the ability of transcription factors to bind. Another possibility is that alteration of the chromatin environment directly affects the kinetics of transcription (*Coulon et al., 2014*) by altering how fast the RNA polymerase elongates.

For the *Arabidopsis* floral repressor gene *FLOWERING LOCUS C* (*FLC*), it has been shown that expression levels are quantitatively reduced by a prolonged duration of cold. This quantitative response is achieved through individual *FLC* gene copies making a *cis*-mediated, digital switch from an 'ON' (expressing) state to an 'OFF' (silenced) state (*Angel et al., 2011*). This switch is asynchronous between gene copies, even in the same cell, with the number switched OFF increasing over time in the cold. This results in a gradual decrease in *FLC* expression over time at a whole plant level with silenced gene copies covered by high levels of the silencing histone mark H3K27me3 controlled by the Polycomb system through Polycomb Repressive Complex 2 (PRC2). This mode of regulation is called 'digital,' to highlight the discrete ON/OFF states for each gene copy (*Figure 1A*, Digital Regulation, *Munsky and Neuert, 2015*). Such digital regulation has also been observed in many other systems, both natural (*Saxton and Rine, 2022*) and engineered (*Bintu et al., 2016*).

An alternative mode of quantitative gene regulation is one that allows graded expression levels to be maintained at each gene copy, rather than just an ON or an OFF state. Quantitative regulation at the cell population level can then be achieved by tuning the expression level uniformly at all gene copies, as in inducible gene expression systems. A well-characterized example of such behavior is in the level of stress-responsive gene expression as controlled by the transcription factor Msn2 in budding yeast (*Stewart-Ornstein et al., 2013*). This graded mode of regulation is called 'analog' (*Figure 1A*, Analog Regulation, ), in contrast to the digital alternative.

While *FLC* loci are known to have digital behavior during and after a cold treatment, allowing them to robustly hold epigenetic memory of cold exposure, it remains unknown how the starting expression levels (prior to cold) are regulated quantitatively in terms of analog versus digital control. After plants germinate, they grow as seedlings that have not experienced any cold exposure, or digital Polycomb switching, so quantitative variation in *FLC* expression seen in young seedlings could represent different cellular proportions of digitally regulated *FLC*, as well as graded transcriptional changes. More generally, elucidating the interplay between analog and digital control is essential for a more in-depth understanding of quantitative gene regulation. Although digital and analog modes of repression have been separately studied in the past (see, e.g., ), how these two fundamentally different modes of regulation might be combined has not been considered. *FLC* is an ideal system to study this question due to its digital control after cold, as well as the wealth of knowledge about its regulation at all stages (*Berry and Dean, 2015*; *Wu et al., 2020*).

*FLC* levels are set during embryogenesis by competition between an *FLC* activator called *FRIGIDA* (*FRI*) and the so-called autonomous repressive pathway (*Figure 1B*, *Li et al., 2018*; *Schon et al., 2021*). The commonly used *Arabidopsis* accessions Ler and Col-0, have mutations in the *FRI* gene, allowing the autonomous pathway to dominate, thereby repressing *FLC* during vegetative development and resulting in a rapidly cycling summer annual lifestyle (*Johanson et al., 2000*). On the other hand, genetic introduction of an active *FRI* allele (Col*FRI*, *Lee and Amasino, 1995*) generates an initially high *FLC* expression state, which then requires cold for *FLC* repression and subsequent flowering. However, both these cases, with high (Col*FRI*) or low (Ler, Col-0) *FLC* expression, are extreme examples, where essentially all *FLC* loci are either expressed to a high level or not, before cold. It is

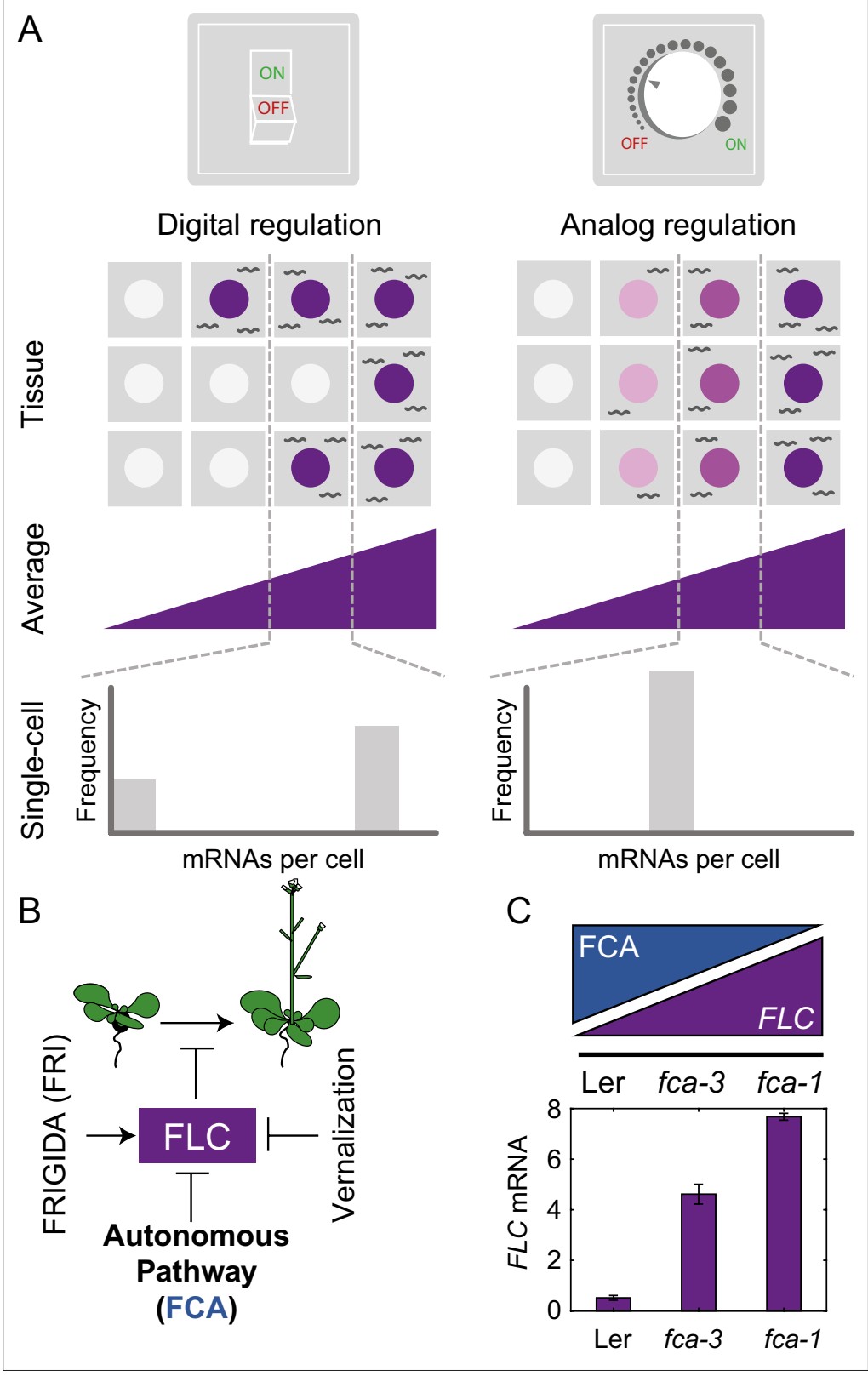

**Figure 1.** Schematic of digital and analog gene regulation. (**A**) Digital regulation (left) corresponds to loci being in an 'ON' state (purple) or 'OFF' state (white), where we assume for simplicity that there is only one gene copy per cell. At the tissue level, moving from low to high average expression (columns left to right) is achieved by a change in the fraction of cells in each of the two states. This mode is distinct from analog regulation (right), where each

*Figure 1 continued on next page*

*Figure 1 continued*

cell has a graded expression level that roughly corresponds to the overall population average. (**B**) *FLC* represses the transition to flowering and is controlled by *FRIGIDA*, the autonomous pathway and the vernalization pathway (inducing digital epigenetic silencing in the cold). (**C**) Reduced *FCA* activity leads to higher cell population-level *FLC* expression in *fca* mutants. Wildtype Ler has lowest *FLC* expression, *fca-3* intermediate and *fca-1* highest. Expression is measured by qPCR relative to the house-keeping gene index (geometric mean of *PP2A* and *UBC*). Error bars show SEM of n = 3 biological replicates measured 7 d after sowing. Statistical tests: multiple comparisons following ANOVA (F-value = 214.62, p-value=$2.6 \cdot 10^{-6}$) with Tukey HSD post hoc tests for *fca-3* – *fca-1*: p-value=0.00029; *fca-3* – Ler: p-value=$5.5 \cdot 10^{-5}$; *fca-1* – Ler: p-value<$2.6 \cdot 10^{-6}$ .

The online version of this article includes the following figure supplement(s) for figure 1:

**Figure supplement 1.** Characterization of *fca* alleles and FLC-Venus transgene.

therefore difficult to use these genotypes to understand any possible interplay between digital and analog control as only the extremes are exhibited. Instead, what is required is a genotype with intermediate *FLC* expression at a cell population level during embryogenesis. Such a genotype at a single gene copy level might exhibit either a graded (analog) or all-or-nothing (digital) *FLC* expression before cold, thereby allowing dissection of whether analog or digital regulation is at work.

In this work, we exploited mutant alleles in *FLOWERING CONTROL LOCUS A* (*FCA*), which is part of the *FLC*-repressive autonomous pathway, thereby systematically varying overall *FLC* levels. One of these mutants, *fca-1* (**Koornneef et al., 1991**), is a complete loss of function and therefore exhibits late flowering (**Figure 1—figure supplement 1A**) and high *FLC* expression before cold (**Figure 1C**, **Figure 1—figure supplement 1B–D**), similar to Col*FRI*. The wildtype, Ler, has a fully functional autonomous pathway and so exhibits low *FLC* levels before cold (**Figure 1C**, **Figure 1—figure supplement 1B–D**) and early flowering (**Figure 1—figure supplement 1A**), with the *FLC* gene covered by the silencing histone mark H3K27me3 (**Wu et al., 2020**). Crucially, however, the *fca-3* and *fca-4* mutants lead to compromised FCA function (see 'Materials and methods' and **Koornneef et al., 1991**). In *fca-3*, there is a splice site mutation that leads to changed exon structure through variable use of alternative splice sites with either loss or misfolding of the C-terminus (**Macknight et al., 2002**). The *fca-4* allele is the result of a large genomic inversion that disrupts the *FCA* gene at the 3' end of exon 4 (**Page et al., 1999**). The expressed 3' fragment contains the second RNA-binding domain and the C-terminal region of the protein including the WW protein interaction domain, sufficient to give an intermediate flowering time phenotype (**Page et al., 1999**). Both mutants display intermediate cell population-level *FLC* expression (**Figure 1C**, **Figure 1—figure supplement 1B–D**) and flowering time (**Figure 1—figure supplement 1A**), indicating partial functionality of the FCA protein in these mutants. Crucially, this intermediate property allows us to systematically dissect the interplay of analog and digital regulation at *FLC* before cold using a combination of single cell and whole plant assays, together with mathematical modeling, revealing a temporal separation between the two regulatory modes.

## Results
### Analysis of *fca* alleles reveals both analog and digital regulation at *FLC*

To investigate the mode of repression arising from regulation by the autonomous pathway, we utilized the *fca-1* and *fca-3* mutants, as well as the parental Ler genotype, and assayed how individual cells varied in their *FLC* expression in these three genotypes at 7 d after sowing. We quantified the number of individual mRNAs per cell using single-molecule fluorescence in situ hybridization (smFISH) (**Figure 2A and B**, **Figure 2—figure supplement 1A and B**). The endogenous Ler *FLC* carries a Mutator transposable element (TE) in intron 1, silencing *FLC* expression (**Liu et al., 2004**). Possibly because of the TE, we observed *FLC* mRNA accumulation in the nucleolus in this background (**Figure 2—figure supplement 1A**). To avoid this complication, we transformed a Venus-tagged *FLC* into Ler (**Figure 2A**, ). The transgenic *FLC* sequence was from Col-0, which does not contain the TE. We then crossed the FLC-Venus into our mutant genotypes (*fca-1, fca-3, fca-4*), thus ensuring that the transgene is in the same genomic location and therefore that changes in its expression will be due to the *FCA* mutations in these lines. We also used smFISH probes agaisnt the Venus and *FLC* sequence. These probes generated a signal specific to the transgenic *FLC* copy in these plants

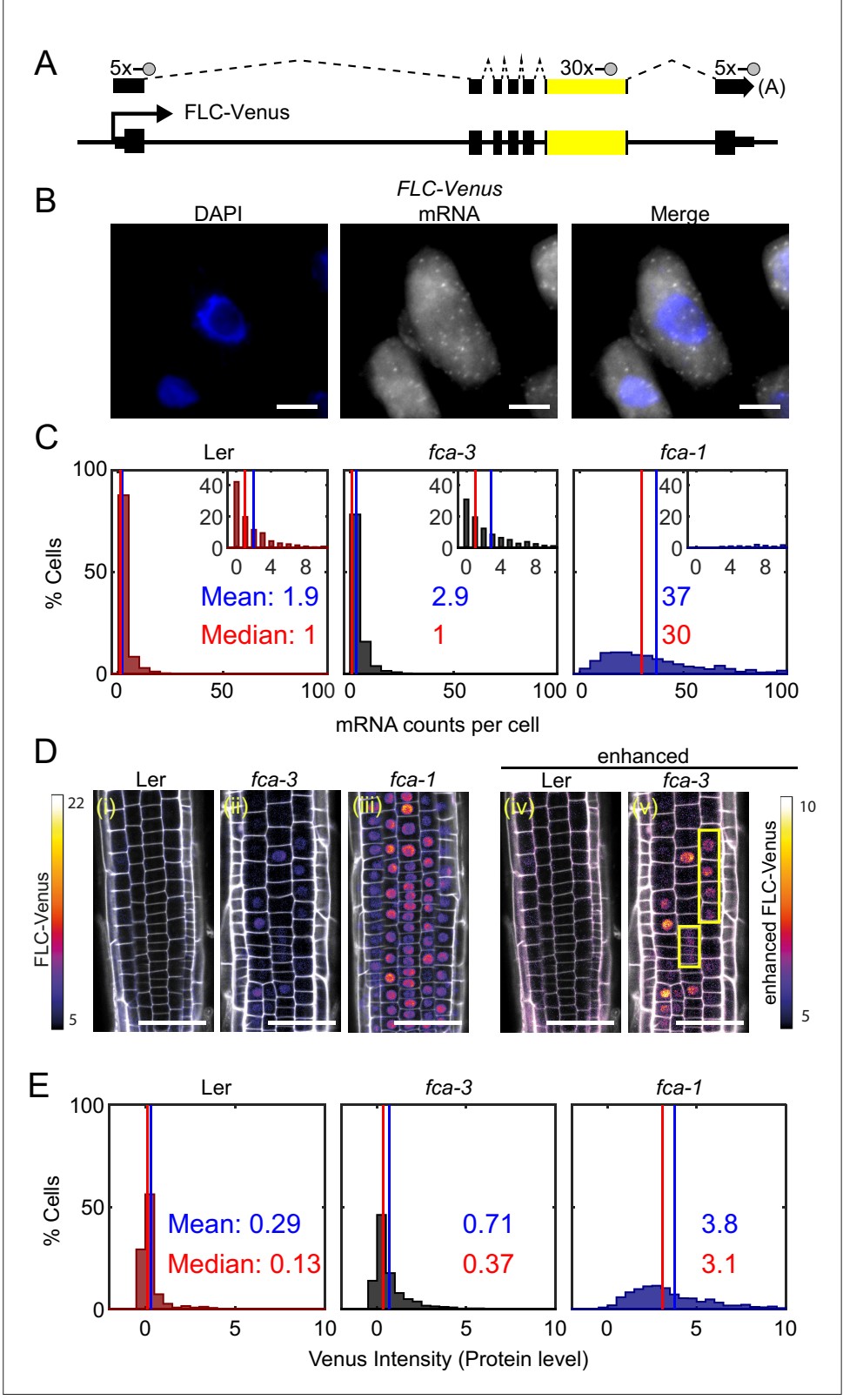

**Figure 2.** *FLC* expression per cell in *fca* mutants. (**A**) Schematic diagram of *FLC-Venus* locus with transcript and exonic probe position indicated. A total of 40 probes were designed, 10 against the *FLC* sequence and 30 for the *Venus* sequence. (**B**) Detection of *FLC-Venus* transcripts in single cells. Representative images of isolated cells with DAPI staining (blue) and *FLC-Venus* mRNA (gray) obtained from *Arabidopsis thaliana* root squashes. Scale

*Figure 2 continued on next page*

*Figure 2 continued*

bar, 5 μm. (**C**) Histograms of single-molecule fluorescence in situ hybridization (smFISH) results for each genotype (Ler, *fca-3*, *fca-1*) showing the number of single-molecule *FLC-Venus* RNAs detected per cell. We show the distribution for Ler, but note that this is likely indistinguishable from background using our methods. The means and medians of the distributions are indicated in each panel. Insets show the same data for mRNA counts between 0 and 10 (where 10 is the 95th percentile of the *fca-3* data). For Ler n = 853, *fca-3* n = 1088, *fca-1* n = 792; data from 7 d after sowing (two independent experiments). Statistical tests: three-way comparison with Kruskal–Wallis ($\chi^2$ (2) = 1611.21, p-value=0) and pairwise comparisons with Wilcoxon rank sum tests with Bonferroni-adjusted p-values for *fca-3* – *fca-1*: adj. p-valu =5.5 · 10$^{-275}$; *fca-3* – Ler: adj. p-value 1.2 · 10$^{-9}$; *fca-1* – Ler: adj. p-value=2.9 · 10$^{-257}$. (**D**) Representative confocal images of roots for each genotype. FLC-Venus intensity indicated by color maps; gray shows the propidium iodide (PI) channel. Same settings were used for imaging and image presentation in (**i–iii**). Images in (**iv**) and (**v**) are the same as (**i**) and (**ii**), respectively, but adjusted to enhance the Venus signal by changing brightness and contrast (please note different scale of color map). Yellow boxes in (**v**) show short files of ON cells. Scale bar, 50 μm. (**E**) Histograms of FLC-Venus intensity per cell in each genotype. The means and medians of the distributions are indicated in each panel. For Ler n = 537 cells from 6 roots, *fca-3* n = 1515 cells from 14 roots, *fca-1* n = 1031 cells from 11 roots; data from 7 d after sowing (two independent experiments). Statistical tests: three-way comparison with Kruskal–Wallis ($\chi^2$ (2) = 1607.56, p-value=0) and pairwise comparisons with Wilcoxon rank sum tests with Bonferroni-adjusted p-values for *fca-3* – *fca-1*: adj. p-value=8.7 · 10$^{-281}$; *fca-3* – Ler: adj. p-value=1.7 · 10$^{-32}$; *fca-1* – Ler: adj. p-value=1.2 · 10$^{-201}$.

The online version of this article includes the following figure supplement(s) for figure 2:

**Figure supplement 1.** Single-molecule fluorescence in situ hybridization (smFISH) method for FLC-Venus imaging.

**Figure supplement 2.** Threshold for ON/OFF state of cells in single-molecule fluorescence in situ hybridization (smFISH) experiments.

**Figure supplement 3.** FLC-Venus imaging in *fca* alleles – root replicates and *fca-4*.

**Figure supplement 4.** FLC-Venus imaging in *fca* mutants and wildtype, in young leaf tissue.

**Figure supplement 5.** Segmentation and quantification method for FLC-Venus protein intensity per nucleus.

---

since we observed no signal in lines without the transgenic *FLC* (***Figure 2—figure supplement 1B***). Comparison of whole-plant gene expression showed similar behavior in the mutants for both the endogenous and transformed *FLC* (***Figure 1—figure supplement 1C and D***). Focusing on the *Venus* sequence conferred the additional advantage that we were able to use the same lines for both mRNA and protein level quantifications (see below).

We observed that the mean number of *FLC* mRNA transcripts per cell was highest in *fca-1*, lowest in Ler, and intermediate in *fca-3*, with the distributions being significantly different (***Figure 2C***, all differences significant with Bonferroni-adjusted p≤1.2 · 10$^{-9}$), with 89% of *fca-1* cells having higher expression than the 95th percentile of *fca-3* cells. Furthermore, when looking only at the ON cells (***Figure 2—figure supplement 1C***; ON cells defined here as cells with more than three mRNAs counted, though the conclusion is unchanged with alternative thresholds; ***Figure 2—figure supplement 2***), the mRNA numbers in *fca-3* were only around 1/5 of those in *fca-1*, much less than one-half, ruling out that the reduced levels in *fca-3* were solely due to silencing of one of the two gene copies. The mRNA distribution of *fca-3* cells is significantly different from Ler, confirming that *fca-3* is not simply all digitally OFF. However, we note that the Ler images contain noisy signals due to background (see 'Materials and methods'), resulting in the detection of ON cells with similar mRNA counts as in *fca-3* (***Figure 2—figure supplement 1***). Despite this, *fca-3* has considerably more ON cells than in Ler (***Figure 2—figure supplement 1***), indicating that counts for *fca-3* are not simply background. Visual inspection and manual counting of images further confirmed these conclusions ('Materials and methods'). Nevertheless, mRNA numbers in many *fca-3* cells were close to zero, suggesting that digital silencing may also be relevant in this case. Hence, the differences in overall expression between the mutant genotypes appeared to have two components. Firstly, there were different fractions of apparently silenced cells (digitally OFF) without any appreciable expression (three or fewer mRNAs counted): 72% in *fca-3* and 1.5% in *fca-1*. Secondly, in cells that did express *FLC* (digitally ON), the *FCA* mutations lead to an analog change in their expression levels, so that there was more *FLC* mRNA in *fca-1* ON cells than in the *fca-3* ON cells (***Figure 2—figure supplement 1C***). This behavior clearly differed from the digital regulation observed at *FLC* after cold treatment (***Figure 1A***, ***Angel et al., 2011***; ***Rosa et al., 2016***).

We additionally used confocal live imaging to investigate FLC protein levels in individual cells in the root tip (**Figure 2D**, **Figure 2—figure supplement 3**) and in young leaves (**Figure 2—figure supplement 4**). We found similar protein expression patterns in roots and leaves, suggesting that each mutation was having similar effects in different tissues. Therefore, conclusions we draw from root experiments can be extended to *FLC* regulation in leaves. Imaging in roots has clear technical advantages, as well as the presence of clonal cell files which can inform heritability, and so further microscopy experiments are in root tissue only. After cell segmentation, we quantified the FLC-Venus intensity within the nuclei, comparing the different genotypes (**Figure 2E**, **Figure 2—figure supplement 5**; all differences are significant with $p$). This procedure allowed us to combine information of relative protein levels with the cell positions in the root. Median intensity levels of FLC-Venus per cell and the overall histogram distribution revealed again intermediate levels of FLC protein in *fca-3*, relative to Ler and *fca-1*. At the same time, there was again strong evidence in favor of a digital component for *FLC* regulation (**Figure 2D(v)**): in cells with the lowest protein levels, these levels were similar in *fca-3* and in Ler, again supporting a digital OFF state. Furthermore, we could see a mix of distinct ON and OFF cells by enhancing the *fca-3* images to increase the Venus intensity to a similar level as in *fca-1*. By contrast, in *fca-1* all cells were ON, whereas in Ler, cells appeared OFF even with an equivalent adjustment (**Figure 2D(iv)**, **Figure 2—figure supplement 3**). We could also infer a potentially heritable component in the ON/OFF states as short files of ON cells could be observed in *fca-3* (**Figure 2D(v)**, **Figure 2—figure supplement 3**, yellow boxes). In *fca-4*, we observed similar short ON files (**Figure 2—figure supplement 3**, yellow boxes), suggesting that this is a general feature that is not specific to *fca-3*. Overall, our results support a combination of analog and digital regulation for *FLC*: in Ler most cells were digitally OFF, in *fca-1* all cells were digitally ON, while in *fca-3* a fraction of cells were OFF, but for those cells that were ON, the level of *FLC* expression was reduced in an analog way relative to *fca-1*.

## *FLC* RNA and protein are degraded quickly relative to the cell cycle duration

Our results strongly pointed toward a digital switching component being important for *FLC* regulation by the autonomous pathway, but with an analog component too for those loci that remain ON. A study in yeast previously reported on the expected RNA distributions for the two cases of analog and digital control (**Goodnight and Rine, 2020**). An important additional consideration when interpreting the analog/digital nature of the regulation concerns the half-lives of the mRNA and protein. In a digital scenario, long half-lives (**Rahni and Birnbaum, 2019**) are expected to broaden histograms of mRNA/protein levels due to the extended times needed for mRNA/protein levels to increase/decrease after state switching. This could lead to a possible misinterpretation of analog regulation, for example, if intermediate levels of mRNA/protein remain in OFF cells that are descended from ON cells. In contrast, short half-lives will lead to clearer bimodality. Furthermore, what might look like a heritable transcription state could also appear due to slow dilution of a stable protein, as observed in other cases (**Kueh et al., 2013**; **Zhao et al., 2020**). We therefore needed to measure the half-lives of the mRNA and protein to interpret our observations appropriately.

*FLC* mRNA has previously been shown to have a half-life of approximately 6 hr (**Ietswaart et al., 2017**) in a different genotype (Col*FRI*) to that used here. We measured the half-lives of both RNA and protein in our highly expressing *FLC* line, *fca-1* (**Figure 3—figure supplement 1**). The RNA half-life measurement used actinomycin D treatment, inhibiting transcription, whereas the protein measurement used cycloheximide, arresting protein synthesis. The half-lives were then extracted from the subsequent decay in mRNA/protein levels. We found that both degradation rates were quite fast, with half-lives of ~5 and ~1.5 hr, respectively, for the mRNA and protein (**Figure 3—figure supplement 1**). These timescales are short compared to the cell cycle duration, here of ~1 d (**Rahni and Birnbaum, 2019**), let alone compared to the timescale of development. Therefore, slow degradation is unlikely to be the cause of the apparent analog regulation and of the observed heritability seen in our root images. Furthermore, the short protein half-life indicates that any potential effects from growth causing dilution, and thus a reduction in protein concentrations, will also be small.

## Switching of *FLC* loci to an OFF state over time in *fca-3*

To understand the nature of potential digital switching, it is important to determine whether switching occurs from ON to OFF, OFF to ON, or in both directions. If most loci are switching one-way only, in either direction, this would lead to a gradual change of overall *FLC* expression over time. Alternatively, two-way switching or non-switching in at least a few cells would be necessary to have a constant concentration of *FLC* mRNA/protein over time. These considerations therefore raised the related question of whether cell population-level silencing is at steady state at the time of observation or whether we are capturing a snapshot of a transient behavior, with cells continuing to switch over developmental time.

To test if *FLC* expression is changing over time, we sampled *FLC* expression in the intermediate *fca-3* mutant, as well as *fca-1* and Ler, at 7, 15, and 21 d after sowing. This experiment revealed a decreasing trend in *fca-3* and Ler (**Figure 3A**), which did not seem to be due to a change in *FCA* expression over the same timescale (**Figure 3B**). This was particularly clear when comparing *FLC* and *FCA* in *fca-3* between 7 and 15 d: since FCA is a repressor of *FLC* expression, a decrease in both suggests that FCA is not the cause of the *FLC* expression decrease. We therefore concluded that the most likely explanation was primarily one-way switching from an otherwise heritable *FLC* ON state, to the heritable silenced OFF state occurring digitally and independently at individual loci in *fca-3* and Ler. We note that Ler was already mostly OFF at the starting time of 7 d, and so the significant but very slow rate of decrease (slope: –0.038, p-value: $4.8 \cdot 10^{-6}$ , **Figure 3A**; slope: –0.023, p-value: 0.0030, **Figure 3—figure supplement 2A**) is interpreted as a small number of remaining ON cells continuing to switch OFF. However, at the later timepoints Ler had transitioned to flowering and so no biological conclusions were drawn from the *FLC* dynamics at those times in this genotype. Rather, this data was used as a negative control for the root imaging experiments below. In *fca-1* a downward trend was not statistically significant (p=0.12) in **Figure 3A**, likely due to the wide error bars at the first timepoint. In another experiment, however (**Figure 3—figure supplement 2A**), there was slow but significant decrease also in *fca-1*, suggesting that there might be some cells switching OFF also in that case, but more slowly compared to *fca-3*.

By imaging FLC-Venus, we observed that the fraction of ON cells was indeed decreasing in *fca-3*, over the time course (**Figure 3C and D**, **Figure 3—figure supplement 3**). In fact, after 21 d the pattern of ON/OFF cells in the *fca-3* roots was very similar to that of plants that had experienced cold leading to partial cell population-level *FLC* shutdown, with the majority of cell files being stably repressed, but still with some long files of cells in which *FLC* was ON (compare timepoint 21 in **Figure 3C** and figures in **Berry et al., 2015**). It remains possible that there is a low level of switching in the opposite direction, from OFF to ON, in *fca-3* (see below). The other genotypes did not show this statistically significant decreasing trend in the FLC-Venus data. In *fca-1*, the fraction of ON cells did not show a consistent pattern, while in Ler, the fraction of ON cells remained essentially constant (at a very low level). These results differed from the qPCR data (see above), suggesting in Ler, for example, that epidermis cells in the root tip could switch off early, while a small number of ON cells remain in other tissues of the plant that continue to switch off, thereby explaining the slow decrease in the qPCR experiment (**Figure 3A**, **Figure 3—figure supplement 2A**).

We also examined whether the change over time in *fca-3* could be due to an analog change in the expression of the ON cells rather than a decreasing number of ON cells. By setting a threshold at an intensity of 1, we look at only the tail of the ON cell distribution (normalized by the total number of these cells in each condition) (**Figure 3—figure supplement 4**). We see that high-intensity cells are still present at later times and do not show a reduction in intensity relative to earlier timepoints. This finding is consistent with digital regulation: a decrease in the number of ON cells, but with all remaining ON cells expressing *FLC* at similarly high levels over time. Overall, our results in *fca-3* are consistent with progressive digital switching of *FLC* loci, primarily in the ON-to-OFF direction.

## Mathematical model incorporating digital switching can recapitulate the *FLC* distribution over time in *fca-3*

We finally developed a mathematical model for *FLC* ON/OFF state and protein levels in the root (**Figure 4A**, **Figure 4—figure supplement 1**, 'Materials and methods') to test if the analog and digital components inferred from the data were sufficient to reproduce the experimentally observed patterns. The model incorporated digital *FLC* state switching embedded in simulated dividing cells

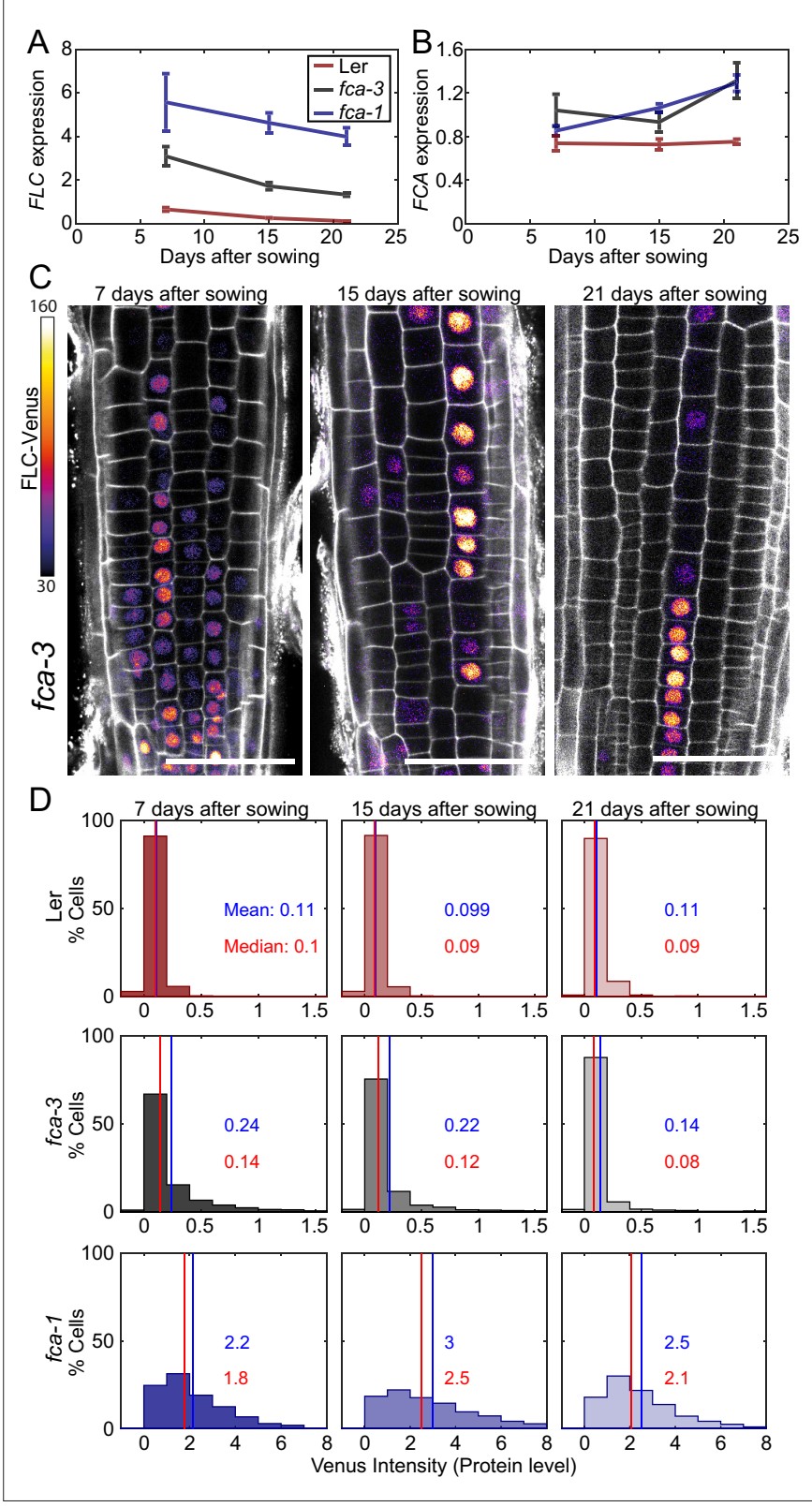

**Figure 3.** Experimental observation of gradual *FLC* silencing. (**A**) Timeseries of *FLC* expression in *fca* mutant alleles transformed with the FLC-Venus construct. Expression is measured by qPCR in whole seedlings relative to the house-keeping gene index (geometric mean of *PP2A* and *UBC*). Error bars show SEM of n = 3 biological replicates. Statistical tests: samples were excluded as outliers based on Grubbs' test with alpha = 0.05. Linear

*Figure 3 continued on next page*

*Figure 3 continued*

regression on timeseries for each genotype. Slope for *fca-3* = −0.13, p-value=$2.0 \cdot 10^{-4}$; slope for *fca-1* = −0.11, p-value=0.12; slope for Ler = −0.038, p-value=$4.8 \cdot 10^{-6}$. (**B**) Timeseries of *FCA* expression, otherwise as in (**A**). Statistical tests: samples were excluded as outliers based on Grubbs' test with alpha = 0.05. Linear regression on timeseries for each genotype. Slope for *fca-3* = 0.021, p-value=0.22; slope for *fca-1* = 0.032, p-value=$3.8 \cdot 10^{-4}$; slope for Ler = $1.1 \cdot 10^{-3}$, p-value=0.84. (**C**) Representative images of *fca-3* roots by confocal microscopy. FLC-Venus intensity indicated by color map; gray shows the PI channel. Same settings were used for imaging and image presentation. Scale bar, 50 μm. (**D**) Histograms of FLC-Venus intensity per cell at each timepoint. The means and medians of the distributions are indicated in each panel. For Ler: 7 d, n = 1121 cells from 10 roots (three independent experiments); 15 d, n = 1311 cells from 9 roots (two independent experiments); 21 d, n = 1679 cells from 12 roots (three independent experiments). For *fca-3*: 7 d, n = 2875 cells from 24 roots (three independent experiments); 15 d, n = 3553 cells from 23 roots (three independent experiments); 21 d, n = 3663 cells from 21 roots (three independent experiments). For *fca-1*: 7 d, n = 1022 cells from 9 roots (three independent experiments); 15 d, n = 1770 cells from 12 roots (three independent experiments); 21 d, n = 2124 cells from 12 roots (three independent experiments). Statistical tests: linear regression on timeseries for each genotype. Slope for *fca-3* = −0.0077, p-value=$4.0 \cdot 10^{-46}$; slope for *fca-1* = 0.018, p-value=0.00064; slope for Ler = -0.00015, p-value=0.44.

The online version of this article includes the following figure supplement(s) for figure 3:

**Figure supplement 1.** Degradation rate of *FLC* mRNA and protein levels.

**Figure supplement 2.** *FLC* and *FCA* expression in whole seedlings over time .

**Figure supplement 3.** FLC-Venus time-course replicates in *fca* alleles.

**Figure supplement 4.** Experimental intensity of FLC-Venus in ON cells does not change over time in *fca-3*.

---

in the root. We allowed the *FLC* state in the model to switch either ON to OFF, or OFF-to-ON. We focused on modeling the switching dynamics and used distributions for the protein levels in cells with two ON loci, one ON and one OFF, and two OFF empirically fitted to our data (***Figure 4A***, 'Materials and methods'). Possible effects from cell size and burstiness are incorporated into these empirical distributions implicitly through this fitting.

In the plant root, there is a mix of dividing and differentiated cells. Our experimental observations capture cells primarily in the division zone of the root, but even within this region, cell cycle times are not the same in all cells (***Rahni and Birnbaum, 2019***). To generate the model cell files, we used cell cycle lengths based on the literature (***Rahni and Birnbaum, 2019***).

With these assumptions, our model could be fit ('Materials and methods') to replicate the observed pattern of increasing OFF cells in *fca-3* roots, as well as the quantitative histograms for protein levels in *fca-3* (***Figure 4B***). In terms of the switching, we found the best fit where the OFF-to-ON rate is over 10 times slower than the ON-to-OFF rate (***Figure 4B***, ***Supplementary file 1***), supporting predominantly one-way switching from an analog into a digitally silenced state. In addition, the model could capture the longer files present in the later timepoints in the data (T21, ***Figure 3C***, ***Figure 4—figure supplement 1***), but unlike in the data these were not more prevalent at later timepoints than earlier. Therefore, the altered prevalence of this effect in the data may suggest additional developmental influence on the heritability of the ON/OFF state at later times in the plant. Overall, however, the model can faithfully recapitulate the developmental dynamics of *FLC* in *fca-3*.

## Discussion

In this work, we have uncovered a combination of analog and digital transcriptional regulation for the gene *FLC*: analog regulation arises through the autonomous pathway, as illustrated by the *fca* mutants, before digital switching into a heritable silenced state. The silenced state is Polycomb dependent, given the similarity of our data to the vernalized state and the role of the Polycomb mark H3K27me3 in silencing *FLC* in wildtype Ler and Col-0 in the absence of cold treatment. In Col-0, in a mutant of the PRC2 component and H3K27 methyltransferase, *clf*, it has also been observed that *FLC* is upregulated and H3K27me3 reduced (***Shu et al., 2019***), though this is only a partial effect due to the presence of an active SWN, another PRC2 H3K27 methyltransferase. Based on previous studies on the interplay between transcription and PRC2 silencing (***Beltran et al., 2016***; ***Berry et al., 2017***; ***Holoch et al., 2021***; ***Lövkvist et al., 2021***), we propose that the rate of digital Polycomb silencing is dependent on the transcription levels. These transcription levels are genetically controlled and

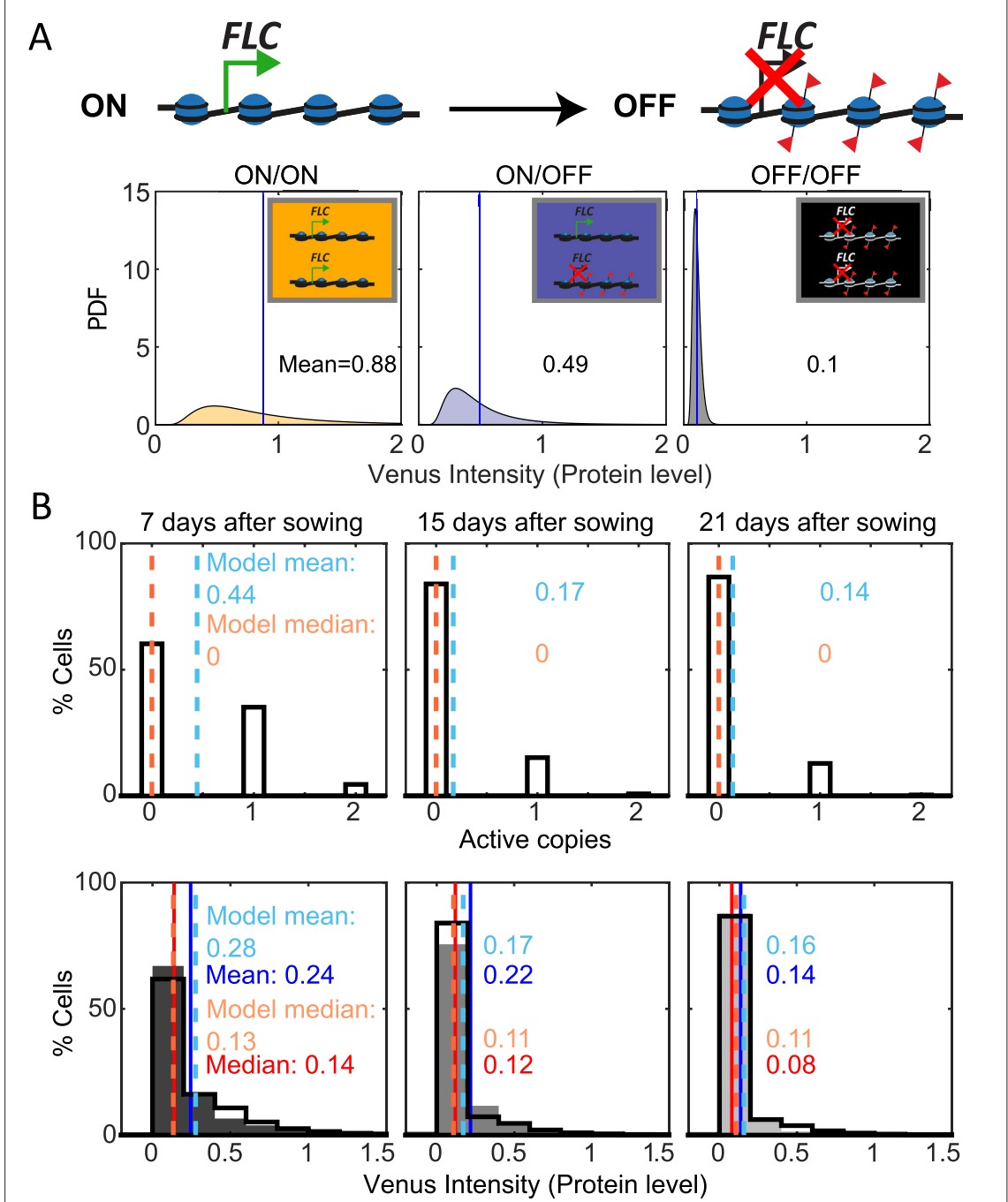

**Figure 4.** Mathematical model captures *FLC* regulation. (**A**) Diagram of mathematical model. Individual *FLC* gene copies can be ON or OFF, such that a cell can be in one of three states depending on the combination of ON and OFF gene copies within it (ON/ON, ON/OFF, OFF/OFF). Venus intensity (corresponding to amount of protein) within a cell was sampled from the distributions shown (described in 'Materials and methods' section), depending on the cell state. The means of the distributions are indicated in each panel. (**B**) Histograms of active *FLC* copies per cell (top) and Venus intensity per cell (bottom) at the indicated timepoints. Model histograms are plotted with black lines around empty bars and experimental data is shown as filled gray histograms with no outline. The means and medians of the distributions are indicated in each panel for the data (solid lines and matching color text) and the model (dashed lines and matching color text). Model simulated 1000 *fca-3* cell files and histograms shown exclude bottom four cells of each file as cells near the QC are also not included in the imaging.

The online version of this article includes the following figure supplement(s) for figure 4:

**Figure supplement 1.** Modeling *FLC* regulation in clonal cell files in the root.

constitute analog regulation since the pathways governing this process are capable of graded expression control. Analog regulation is set within each genotype, as illustrated by the *fca* mutants, before possible digital switching at each gene copy into a heritable silenced OFF state.

In the case of Ler, low analog transcription is not sufficient to significantly oppose silencing and so the switch occurs before our first experimental measurements at 7 d. In *fca-1*, the switch does not occur at all, prevented by high analog transcription in that genotype. However, for *fca-3*, with intermediate analog *FLC* expression, the digital switching occurs slowly and so can be observed in our timeseries experiments, underlining a clear temporal separation of analog and digital transcriptional control. In this way, both analog and digital regulation are combined at *FLC*, with the timescale for switching between these states controlled by the strength of initial analog transcription. In *fca-3*, such switching to the OFF state causes a gradual reduction in *FLC* expression at the whole plant level. Furthermore, we emphasize that there may be additional developmentally regulated processes occurring at *FLC*, in addition to the constant rate of switching. This possibility is underscored by the model not completely recapitulating the long cell files we observe experimentally at the 21-day timepoint.

A prolonged environmental cold signal can also result in a switch from cells expressing *FLC* to nonexpressing OFF cells, a process that occurs independently at each gene copy of *FLC* (**Berry et al., 2015**). These ON/OFF states are then maintained through cell divisions leading to epigenetic memory. This cold-dependent silencing shows only digital characteristics, with highly *FLC*-expressing cells present both before and after intermediate periods of cold, albeit in a lower proportion after. Here, we have addressed how *FLC* is regulated in the absence of cold and show that a similar mode of regulation occurs. Furthermore, at least in the case of *fca-3* with initially intermediate transcription levels, silencing at the cell population level also happens gradually over many cell cycles, like the behavior over prolonged periods of cold. Moreover, images of roots at 21 d after sowing without cold, with fluorescently labeled FLC, are visually strikingly like images of vernalized roots, where in both cases we find whole files of either all ON or all OFF cells. These comparisons suggest similarities in the molecular characteristics of the Polycomb silenced state at *FLC* generated either with or without cold treatment. We hypothesize that in the absence of active Polycomb the *fca* alleles would display only analog differences, without the possibility of digital silencing.

In this work, we have focused on the early regulation of *FLC* by analog and digital pathways, rather than the effect of these on flowering. However, natural variation at *FLC* largely affects expression as the plants are germinating in autumn (**Hepworth et al., 2020**). The analog component of transcriptional regulation could therefore have widespread ecological relevance. For example, the high *FLC* genotype Lov-1, a natural accession from Sweden, has higher *FLC* expression upon germination in early autumn compared to Col*FRI* (**Coustham et al., 2012**). The analog component could be dominating in Lov-1, with all cells having a higher ON level. Consistent with this, Lov-1 also has lower levels of the Polycomb mark H3K27me3 at the *FLC* locus, with potentially few or no OFF cellsn the warm (**Qüesta et al., 2020**).

The main antagonist of the autonomous pathway is the *FRIGIDA* gene, a transcriptional activator of *FLC* (**Figure 1B**), which was not present in the lines used in this study. In fact, it is the antagonism between FRI and the autonomous pathway that determines the starting *FLC* levels, and so we would expect that FRI controls the analog component directly and digital component indirectly. For future work, it would therefore be interesting to investigate transcriptional responses in an *FRI* allelic series. This could be particularly important because natural accessions show mutations in *FRI* rather than in autonomous pathway components, possibly because the autonomous components regulate many more target genes.

Overall, our work has revealed a combination of both analog and digital modes of regulation at *Arabidopsis FLC* before cold, with analog preceding digital. We further propose it is the strength of the initial analog, autonomous pathway transcriptional level that controls the timescale for the switch into the subsequent digital, Polycomb silenced state. Future work is needed to address how this integrated analog and digital regulation affects flowering time, including further exploration of this pathway in leaves and the use of cell-tracking to capture the switching process in action.

## Materials and methods

### Plant material

#### *fca* alleles

*fca-3* and *fca-4* are the result of X-ray mutagenesis, while *fca-1* was induced by EMS (*Koornneef et al., 1991*). For Ler, *fca-3* and *fca-1*, the transcript levels of *FCA* determined with a primer pair binding between exons 16 and 19 are comparable (*Figure 3B*). The primer pair used is specific to the gamma isoform in a wildtype context, which so far has been the only FCA isoform known to generate a functional protein (*Macknight et al., 2002*). The *fca-1* C-T point mutation generates an in-frame early stop codon (*Macknight et al., 1997*), whereas *fca-3* is a splice site mutation from G to A at the 3′ splice site of intron 6 (Chr4:9,210,672). The closest predicted weak alternative splice site (Chr4:9,210,582; 90 bp) is within intron 7, the next constitutive acceptor splice site of intron 7 (Chr4:9,210,518; 154 bp downstream). This suggests that either intron 6 is not spliced out or exon 7 is skipped. Using a C-terminal polyclonal serum antibody, no protein was detected as previously shown in either *fca-3* or *fca-1* (*Macknight et al., 2002*). However, more characterization is needed to determine the exact protein that is made in *fca-3*. *fca-4* contains a breakpoint within intron 4 of *FCA* leading to a chromosomal rearrangement, with the 3′ fragment of *FCA* fused to another gene, leading to partial protein (*Page et al., 1999*). *fca-3* and *fca-4* were empirically selected as intermediate mutants for this study based on their effects on *FLC* levels and on flowering time (*Figure 1C*, *Figure 1—figure supplement 1*).

### Generation of FLC-Venus transgenic lines

pSLJ-755I6 FLC-Venus contains a 12.7 kb genomic fragment containing *FLC* from the Col-0 accession with the Venus coding sequence inserted into the NheI site of *FLC* exon 6, as previously described (*Berry et al., 2015*). This *FLC-Venus* construct was transformed into Ler using *Agrobacterium tumefaciens* and single-copy transgenic lines were selected. We crossed plants that were homozygous for FLC-Venus with the mutant genotypes (*fca-1*, *fca-3*, and *fca-4*) to obtain F2 plants homozygous for both *FLC-Venus* and *fca-1*, *fca-3*, or *fca-4*, by PCR-based genotyping (primers in *Supplementary file 2*) and copy number analysis. *fca-4* genotyping was performed using specific forward primers for the *fca-4* mutation (600 bp) and WT (720 bp). The *fca-3* mutation introduces a recognition sequence for the restriction enzyme BglII, generating fragments of 87 bp and 195 bp (*fca-3*) compared to 282 bp (WT). The *fca-1* mutation generates an additional restriction enzyme recognition site for MseI generating fragments of 157 bp, 125 bp, 18 bp (*fca-1*), and 175 bp, 125 bp for WT. F3 plants derived from these were used for all experiments. We verified by qPCR that *FLC-Venus* showed similar changes in expression as endogenous *FLC* in the *fca-1*, *fca-3*, and *fca-4* backgrounds (*Figure 1—figure supplement 1*).

### Plant growth

Seeds were surface sterilized in 5% v/v sodium hypochlorite for 5 min and rinsed three times in sterile distilled water. Seeds were stratified for 2 d at 4°C in Petri dishes containing MS media without glucose. The plates were placed vertically in a growth cabinet (16 hr light, 22°C) for 1 wk.

### Gene expression analysis

For gene expression timeseries, 20+ whole seedlings were harvested at each timepoint (7-, 15-, and 21-day-old plants). Plant material was snap frozen with liquid nitrogen, ground, and RNA was extracted using the phenol:chloroform:isoamyl alcohol (25:24:1) protocol (*Yang et al., 2014*). RNA was purified with the TurboDNase (Ambion) kit to remove DNA contamination and reverse transcribed into cDNA using SuperScript IV Reverse transcriptase (Invitrogen) and RT primers for genes of interest. Gene expression was measured by qPCR, and data was normalized to PP2A and UBC, unless specified otherwise. Primer sequences are summarized in *Supplementary file 2*.

   *Primer pairs*:

> FLC spliced: FLC_spliced_F / FLC_spliced_R
> FLC unspliced: SDB_FLC_4548_F / SDB_FLC_4701_R
> FLC-Venus spliced: FLC_spliced_F / SDB_FLC-VENUS_ex6_cDNA_744_R
> FLC-Venus unspliced: SDB_FLC_4548_F / SDB_FLC-VENUS_ex6_cDNA_744_R

## Actinomycin D treatment

For actinomycin D (ActD) experiments, 6-day-old plants were initially germinated in non-supplemented media and were transferred to new plates containing ActD. The plants were kept in darkness during ActD treatment. Stock solution of ActD (1 mg/mL dissolved in DMSO) was added to molten MS media to a final concentration of 20 µg/mL. ActD was obtained from Sigma (Cat# A4262-2MG).

## Cycloheximide (CHX) treatment

FLC-Venus protein stability was assayed with the de novo protein synthesis inhibitor cycloheximide (C1988, Sigma-Aldrich) following the procedures in *Zhao et al., 2020*. Briefly, 7-day-old *fca-1* seedlings carrying the *FLC-Venus* transgene were treated in liquid MS medium containing 100 µM CHX. The seedlings were sampled after 0, 1.5, 3, 6, 12, and 24 hr of treatment. A nontreatment control is also included, in which seedlings were soaked in liquid MS medium without the inhibitor CHX for 24 hr. Approximately 1.0 ng seedlings were ground to a fine powder with Geno/Grinder. Total protein was extracted with 1.5 mL buffer (50 mM Tris-Cl pH 8.0, 15 4 mM NaCl, 5 mM MgCl₂, 10% glycerol, 0.3% NP-40, 1% Triton-100, 5 mM DTT, and protease inhibitor). Then each sample was cleaned by centrifugation at 16,000 × g at 4°C for 15 min, 50 µL total protein was taken as input before enrichment with magnetic GFP-trap beads (GTMA-20, Chromo Tek). The input samples were run on a separate gel and used as a processing loading control for the starting level for each sample. The enriched FLC-Venus protein was detected by western blot assay with the antibody anti-GFP (11814460001, Roche). Signals were visualized with chemiluminescence (34095, Pierce) with a secondary antibody conjugated to horseradish peroxidase (NXA931V, GE Healthcare). The chemiluminescence signal was obtained by the FUJI Medical X-ray film (4741019289, FUJI). Quantification was performed with ImageJ after the films were scanned with a printer scanner (RICOH). Ponceau staining was performed with commercial Ponceau buffer (P7170, Sigma-Aldrich) and used as the processing controls. All of the western blot assays were performed with equal weight of whole seedlings.

## smFISH

smFISH was carried out on root squashes as described by *Duncan et al., 2017*. Briefly, root tips from 7-day-old seedlings were cut using a razor blade and placed into glass wells containing 4% paraformaldehyde and fixed for 30 min. Roots were then removed from the fixative and washed twice with nuclease free 1× PBS. Several roots were then arranged on a microscope slide and squashed between the slide and coverslip. This procedure is required to generate a monolayer of cells, thereby decreasing the background, which is necessary to detect the intrinsically low signals from single molecules of mRNA (*Duncan et al., 2016*). Slides were submerged (together with the coverslips) for a few seconds in liquid nitrogen until frozen. The coverslips were then removed, and the roots were left to dry at room temperature for 30 min.

Tissue permeabilization and clearing were achieved by immersing sequentially the samples in 100% methanol for 1 hr, 100% ethanol for 1 hr, and 70% ethanol for a minimum of 1 hr. The ethanol was left to evaporate at room temperature for 5 min and slides were then washed with Stellaris RNA FISH Wash Buffer A (Biosearch Technologies; Cat# SMF-WA1-60). 100 µL of hybridization solution (containing 10% dextran sulfate, 2× SSC, and 10% formamide), with each probe set at a final concentration of 125 nM, was then added to each slide. The slides were left to hybridize at 37°C overnight in the dark.

The hybridization solution containing unbound probes was pipetted out the following morning. Each sample was then washed twice with Stellaris RNA FISH Wash Buffer B (Biosearch Technologies; Cat# SMF-WB1-20) with the second wash left to incubate for 30 min at 37°C. 100 µL of the nuclear stain DAPI (100 ng/mL) was then added to each slide and left to incubate at 37°C for 10 minutes. Slides were then quickly washed with 2× SSC. 100 µL GLOX buffer minus enzymes (0.4% glucose in 10 mM Tris, 2× SSC) was added to the slides and left to equilibrate for 2 min. Finally, this was removed and replaced with 100 µL of GLOX buffer (containing 1 µL of each of the enzymes glucose oxidase [#G0543 from Sigma] and catalase [#C3155 from Sigma]). The samples were then covered by 22 mm × 22 mm No. 1,5 coverslips (VWR), sealed with nail varnish, and immediately imaged.

## smFISH probe synthesis

We used the online program Stellaris Probe Designer version 2.0 from Biosearch Technologies to design probe sequences for *FLC-Venus*. For probe sequences, see **Supplementary file 3** for *FLC-Venus*. For unspliced *PP2A* and for *FLC* exonic, see *Duncan et al., 2016* (reproduced in **Supplementary file 4**).

## Image acquisition

The smFISH slides were imaged using a Zeiss LSM800 inverted microscope, with a 63x water-immersion objective (1.20 NA) and Microscopy Camera Axiocam 503 mono. The following wavelengths were used for fluorescence detection: for probes labeled with Quasar570, an excitation filter 533–558 nm was used and signal was detected at 570–640 nm; for probes labeled with Quasar670, an excitation filter 625–655 nm was used and signal was detected at 665–715 nm; for DAPI, an excitation filter 335–383 nm was used and signal was detected at 420–470 nm; for GFP, an excitation filter 450–490 nm was used and signal was detected at 500–550 nm.

For the FLC-Venus protein level quantification of root samples, optical sections were collected with a Zeiss LSM780 microscope equipped with a Channel Spectral GaAsP detector with a 20x objective (0.8 NA). For z-stacks, the step size was 0.5 µm with a pinhole aperture of 1.5 AU. The overall Z size varied between 45 and 75 slices depending on the orientation of the root. Roots from FLC-Venus lines were immersed in 1 µg/mL propidium iodide (PI, Sigma-Aldrich, P4864) to label the cell wall. For visualization of roots stained with PI, an excitation line of 514 nm was used, and signal was detected at wavelengths of 611–656 nm. For observation of Venus signal, we used a 514 nm excitation line and detected from 518 to 535 nm. To allow comparison between treatments, the same laser power and detector settings were used for all FLC-Venus images.

In FLC-Venus time-course imaging of epidermal root meristems, the Leica SP8X or Stellaris 8 were used with 20x multi-immersion objective (0.75 NA). The Argon (SP8X) or OPSL 514 (Stellaris 8) lasers were used at 5% to excite FLC-Venus and PI at a 514 nm wavelength in bidirectional mode (PI signal was used for set-up). Venus was detected between 518 and 550 nm with the HyD SMD2 detector in photon counting mode; PI was detected at 600–675 nm. Epidermal images were obtained in photon-counting mode with laser speed 200, line accumulation of 6 (pixel dwell time of 2.43 µs), and a Z-step size of 0.95 µm and a pinhole size of 1 AU. For representative images, these were projected such that one single middle slice from the PI channel was used to show the cell outline, onto which 10 slices of FLC-Venus channel were average intensity projected (T7, LSM780 imaging, *Figure 2*) or sum projected for time-course imaging (T7/T15/T21, SP8X imaging, *Figure 3*). The dynamic range of the FLC-Venus signal was pushed from 0 to 255 to 5–22 (for LSM780 images, *Figure 2*) and 30–255 (for SP8X images, *Figure 3*) for all images apart from where enhanced. In enhanced *fca-3*/Ler images, the dynamic range was further pushed to 5–10 (for LSM780 images, *Figure 2*) and to 30–160 (for SP8X images, *Figure 3*) to obtain a similarly strong signal as observed in *fca-1*.

For *fca-4* FLC-Venus imaging, the Zeiss LSM880 was used with a 25x multi-immersion objective (0.8 NA). The Argon laser at 514 nm and 3% was used for excitation of FLC-Venus and PI, with a detection range between 520–550 nm and 600–650 nm, respectively. Z-stacks were taken covering the epidermis, with a step size of 1 µm. In the top two images, the signal was line-averaged four times; for the bottom image, no averaging was performed. For representative images (*Figure 2—figure supplement 3*), five slices of FLC-Venus covering the nucleus were average projected and merged to the middle PI slice. FLC-Venus signal was enhanced to a similar intensity level as in *fca-1* by shifting the dynamic range of the FLC-Venus signal from 0 to 255 to 16–120.

FLC-Venus imaging of young leaves was performed on 9-day-old seedlings. Seedlings were dissected by removing the cotyledons and keeping only the young leaves and shoot meristem. Young leaves were fixed in 4% paraformaldehyde at 4°C overnight. The samples were then treated with 100% methanol and 100% ethanol twice for 15 min each. ClearSee (*Kurihara et al., 2015*) solution was then used to clear the samples for 1 wk. Next, two washes were carried out in 1× PBS. Samples were embedded in a hydrogel according to *Gordillo et al., 2020*. After rinsing with 1× PBS, samples were stained with Renaissance 2200 solution for 15 mins at room temperature. Slides were mounted in vectashield. FLC-Venus levels of the leaf images were quantified manually using ImageJ. The mean fluorescence intensity of a circular region covering the nucleus in each cell was measured, and the mean background of each image was subtracted.

## Image analysis

### FISH analysis

Quantification of FISH probes took place in two stages:

1. Identification of probe locations in the whole 3D image, excluding the top and bottom z-slice from each z-stack due to light reflection at the plant cell wall.
2. Assignment of identified probes to specific cells via segmentation of the image into regions.

To detect probes, a white tophat filter was applied to the probe channel, followed by image normalization and thresholding. Individual probes were then identified by connected component segmentation. The centroids of each segmented region were assigned as the probe's location.

Images were manually annotated with markers to indicate positions of nuclei and whether cells were fully visible or occluded. A combination of the visibility markers and nuclei were used to seed a watershed segmentation of the image, thus dividing the image into regions representing individual cells. Probes within each region were counted to generate the per-cell counts. Occluded cells were excluded from the analysis (probe counts for those regions were ignored). Segmentation and probe detection parameters were optimized, with different thresholds used for different replicates and genotypes. This was necessary due to the difference in signal and background of the probes in each of these cases.

Visual inspection of smFISH images revealed high rates of false positive detection in Ler, likely due to very low levels of expression. In the absence of real signals, unspecific signals can be counted through our automated quantification pipeline as a consequence of unspecific probe binding or general background. When expression levels are high, this effect is negligible, but when expression levels are very low, it can become significant. Therefore, differences at the cellular mRNA level between *fca-3* and Ler are likely higher than the ones presented in *Figure 2C*. The similar count range observed in Ler and *fca-3* 'ON' cells (*Figure 2—figure supplements 1C and 2*) suggests that there is an overlap between the distribution of real ON cells in *fca-3* and background signal (in both Ler and *fca-3*). However, the higher frequency of such cells in *fca-3* compared to Ler confirms that these are not simply the result of background measurements. This is reflected in the significantly different distribution of Ler and *fca-3* cells in *Figure 2C*. Furthermore, visual inspection of the images confirmed that in *fca-3* we can confidently detect real signals (higher than in Ler and lower than *fca-1*), confirming the analog component.

Custom code is available at https://github.com/JIC-Image-Analysis/fishtools (copy archived at *Hartley and Antoniou-Kourounioti, 2022a*).

### FLC-Venus fluorescence intensity in roots

To measure per-cell FLC-Venus intensities in roots, we developed a custom image analysis pipeline to extract cell structure information from the PI cell wall information and use this to measure per-cell nuclear fluorescence. Images were initially segmented using the SimpleITK (*Beare and Lehmann, 2006*) implementation of the morphological watershed algorithm. Reconstructed regions touching the image boundaries and those below a certain size threshold were removed. Segmentations were then manually curated to merge over-segmented regions and assign file identities to the resulting segmented cells. This curation was performed using custom software, able to merge segmented cells, resegment cells from specified seeds, and split cells along user-defined planes. To approximate the nuclear position, we fitted a fixed size spherical volume of 15 voxels radius to the point of maximal FLC-Venus intensity in each reconstructed cell (*Figure 2—figure supplement 5*). Cells where the sphere overlap fraction (between sphere and cell) was less than 55% were excluded from the analysis. Per-voxel mean FLC-Venus intensities for each cell were then calculated by dividing the summed intensity within the intersection of the spherical region and the cell, by the fixed sphere volume. A background correction was performed by the following calculation to estimate the FLC-Venus intensity in the nucleus ('sphere_filled,' also see *Supplementary file 5*): sphere_filled = (mean_in_sphere / overlap_fraction) - mean_outside_sphere.

The 'mean_in_sphere' intensity was first divided by the 'overlap_fraction,' the fraction of the sphere which overlaps the cell, to correct for regions of the sphere outside the cell. The mean intensity of the region outside the sphere ('mean_outside_sphere'), which is the background signal, was then subtracted from this overlap-corrected sphere intensity. This background subtraction is also the cause

for the negative values in a fraction of OFF cells where the signal inside and outside the sphere are both background noise. Because of the sphere's prescribed size, in some cases even the brightest sphere inside a cell will have lower intensity than the remaining cell (e.g., by necessarily including the darkest region of the cell), leading to these negative values.

Custom code for initial segmentation and Venus intensity measurement is written in the Python language (*Python Software Foundation, 2023*), and is available at https://github.com/JIC-Image-Analysis/root_measurement (copy archived at *Hartley and Antoniou-Kourounioti, 2022b*).

The code for the segmentation curation software is available at https://github.com/jfozard/segcorrect (copy archived at *Fozard, 2022*).

## Mathematical model

We constructed a model for *FLC* chromatin states and protein levels in the root (*Figure 4A*) and used it to simulate root cell files and generate simulated protein-level histograms (*Figure 4B*, *Figure 4—figure supplement 1*).

The root was represented as a collection of cell files (such as the 12 representative cell files shown in *Figure 4—figure supplement 1*), and 1000 cell files were simulated in total. Each cell file consisted of a list of 30 cells (*Supplementary file 1*), ordered from the root tip toward the rest of the plant. This region was intended to approximately match the division zone and imaging region. Cells were able to divide, giving rise to two daughter cells and pushing cells that were further from the tip upward. Cells escaping the 30 cell limit were removed from the simulation. Cells in a given file were clonally related to each other, and eventually all originated from divisions of the 'initial cells' (the first cells of the cell file, which are adjacent to the quiescent center [QC] of the root apical meristem). Each cell was described by its index along the cell file, its digital chromatin state, its protein concentration (expressed as the Venus intensity to match experimental observations), and its remaining cell cycle duration.

With regard to the digital chromatin state, a cell could be in an ON/ON (both *FLC* copies in the active chromatin state), ON/OFF (one active and one inactive), or OFF/OFF state (both inactive). All cells at the start of the simulation at 0 d are in the ON/ON configuration. The model also included a digital switching process between these cell states. Based on our data, we expected switching to occur at least primarily from a heritable ON state to a heritable OFF state. However, we also simulated switching from OFF to ON in the model to test if we could explain the data also in that case. At each timestep and depending on its digital chromatin state, a single *FLC* copy could switch from an ON to an OFF state with probability $p_{OFF}$, or from an OFF to an ON state with probability $p_{ON}$ (values in *Supplementary file 1*).

In order to compare the simulated cell states against the FLC-Venus data, we processed the model outputs using an additional step. This step gave each ON *FLC* copy an associated protein level, which was sampled from a log-normal distribution with parameters: $\mu_{ON}, \sigma_{ON}^2$. The background within each cell was sampled from a log-normal distribution with parameters: $\mu_{OFF}, \sigma_{OFF}^2$. These four parameters and the switching probabilities $p_{OFF}$ and $p_{ON}$ were manually fitted to the experimental Venus distributions for *fca-3* at 7, 15, and 21 d (*Figures 3D and 4*, *Supplementary file 1*) by adjusting the parameters and visually inspecting until the model fits were judged to be satisfactory. Protein levels for each cell, according to the combination of ON and OFF *FLC* copies (*Figure 4A*), were given by appropriate combinations of random variables, specifically,

- for ON/ON cells: $2e^{X_{ON}} + e^{X_{OFF}}$ (cell background and signal from two ON copies);
- for ON/OFF cells: $e^{X_{ON}} + e^{X_{OFF}}$ (cell background and signal from one ON copy); and
- for OFF/OFF cells: $e^{X_{OFF}}$ (only the cell background),

where $X_{ON} \sim N\left(\mu_{ON}, \sigma_{ON}^2\right)$ and $X_{OFF} \sim N\left(\mu_{OFF}, \sigma_{OFF}^2\right)$. The noisy nucleus background signal ($X_{OFF}$) was assumed to be higher than any real signal originating from OFF *FLC* copies. The log-normal distribution was selected for simplicity and automatic non-negativity, and we found empirically that it gave reasonable fits.

The duration of the cell cycle of each cell was determined by the cell's position along the cell file at the time of the division that created it. These cell cycle times according to position were based on the literature (*Rahni and Birnbaum, 2019*). We used a truncated normal distribution with the mean and standard deviation matching the measured values for epidermis cells. A minimum value was set such

that the cell cycle could not be shorter than 13 hr (the lowest value observed for epidermis or cortex cells in *Rahni and Birnbaum, 2019*).

The simulation process was as follows in each cell file, at each timestep:

Division:

- Remaining cell cycle duration is reduced by timestep in all cells.
- Cells with ≤ 0 remaining cell cycle duration divide, such that the positional index of cells with higher index than the dividing cells is increased by 1.
- For each dividing cell, the two daughter cells (dividing cell and cell with positional index +1 relative to it) are assigned new cell cycle durations according to their position.
- Daughter cells are assigned the same digital chromatin state as the mother cell.
- Cells with positional index greater than 30 are removed from the simulation.

Switching:

- For each cell in the cell file, a random number ($r_i$) is generated, which is compared to the switching probability as followsL
- If the cell is in the OFF/OFF state:
  - With probability $p_{ON}^2$ it will switch to the ON/ON state – if $r_i < p_{ON}^2$
  - With probability $2p_{ON}(1 - p_{ON})$ it will switch to the ON/OFF state – if $r_i \geq p_{ON}^2$ and $r_i < \left(p_{ON}^2 + 2p_{ON}(1 - p_{ON})\right)$
  - With probability $(1 - p_{ON})^2$ it will stay in the OFF/OFF state – if $r_i \geq \left(p_{ON}^2 + 2p_{ON}(1 - p_{ON})\right)$
- If the cell is in the ON/OFF state:
  - With probability $p_{OFF}(1 - p_{ON})$ it will switch to the OFF/OFF state – if $r_i < (p_{OFF}(1 - p_{ON}))$
  - With probability $p_{ON}(1 - p_{OFF})$ it will switch to the ON/ON state – if $r_i \geq (p_{OFF}(1 - p_{ON}))$ and $r_i < (p_{OFF}(1 - p_{ON}) + p_{ON}(1 - p_{OFF}))$
  - With probability $(p_{ON}p_{OFF} + (1 - p_{ON})(1 - p_{OFF}))$ it will stay in the ON/OFF state – if $r_i \geq (p_{OFF}(1 - p_{ON}) + p_{ON}(1 - p_{OFF}))$
- If the cell is in the ON/ON state:
  - With probability $p_{OFF}^2$ it will switch to the OFF/OFF state – if $r_i < p_{OFF}^2$
  - With probability $2p_{OFF}(1 - p_{OFF})$ it will switch to the ON/OFF state – if $r_i \geq p_{OFF}^2$ and $r_i < \left(p_{OFF}^2 + 2p_{OFF}(1 - p_{OFF})\right)$
  - With probability $(1 - p_{OFF})^2$ it will stay in the ON/ON state – if $r_i \geq \left(p_{OFF}^2 + 2p_{OFF}(1 - p_{OFF})\right)$

Custom code is available at https://github.com/ReaAntKour/fca_alleles_root_model (copy archived at *Antoniou-Kourounioti, 2023*).

## Acknowledgements

We thank all members of the Dean, Rosa and Howard groups for excellent discussions. Special thanks to Dr. Cecilia Lövkvist for project coordination. Additionally, we would like to thank all the funders listed below.

## Additional information

### Funding

| Funder | Grant reference number | Author |
| --- | --- | --- |
| Biotechnology and Biological Sciences Research Council | BB/P020380/1 | Caroline Dean Caroline Dean |
| Vetenskapsrådet | 2018-04101 | Anis Meschichi Anis Meschichi |
| Biotechnology and Biological Sciences Research Council | BB/P013511/1 | Caroline Dean Caroline Dean |

| Funder | Grant reference number | Author |
| --- | --- | --- |
| HORIZON EUROPE Marie Sklodowska-Curie Actions | 813282 | Svenja Reeck<br>Svenja Reeck |
| HORIZON EUROPE Marie Sklodowska-Curie Actions | MSCA-IF 101032710 | Lihua Zhao |

The funders had no role in study design, data collection and interpretation, or the decision to submit the work for publication.

## Author contributions

Rea L Antoniou-Kourounioti, Conceptualization, Data curation, Formal analysis, Investigation, Methodology, Writing – original draft, Writing – review and editing; Anis Meschichi, Formal analysis, Investigation, Methodology, Writing – original draft; Svenja Reeck, Data curation, Formal analysis, Investigation, Methodology, Writing – original draft, Writing – review and editing; Scott Berry, Conceptualization, Investigation, Writing – review and editing; Govind Menon, Validation, Investigation; Yusheng Zhao, Resources, Investigation, Methodology; John Fozard, Formal analysis, Software; Terri Holmes, Huamei Wang, Investigation; Lihua Zhao, Methodology; Matthew Hartley, Conceptualization, Resources, Software, Formal analysis, Validation, Investigation, Writing – review and editing; Caroline Dean, Conceptualization, Resources, Supervision, Funding acquisition, Investigation, Writing – original draft, Writing – review and editing; Stefanie Rosa, Conceptualization, Formal analysis, Supervision, Funding acquisition, Methodology, Writing – original draft, Project administration, Writing – review and editing; Martin Howard, Conceptualization, Resources, Formal analysis, Supervision, Funding acquisition, Writing – original draft, Project administration, Writing – review and editing

## Author ORCIDs

Rea L Antoniou-Kourounioti http://orcid.org/0000-0001-5226-521X
Anis Meschichi http://orcid.org/0000-0001-8946-6023
Svenja Reeck https://orcid.org/0000-0002-6362-5310
Scott Berry https://orcid.org/0000-0002-1838-4976
Govind Menon https://orcid.org/0000-0002-1028-5463
Yusheng Zhao https://orcid.org/0000-0002-5893-504X
John Fozard https://orcid.org/0000-0001-9181-8083
Terri Holmes https://orcid.org/0000-0002-8480-0755
Lihua Zhao https://orcid.org/0000-0002-1758-9873
Matthew Hartley https://orcid.org/0000-0001-6178-2884
Caroline Dean https://orcid.org/0000-0002-6555-3525
Stefanie Rosa https://orcid.org/0000-0002-8100-1253
Martin Howard https://orcid.org/0000-0001-7670-0781

## Decision letter and Author response

Decision letter https://doi.org/10.7554/eLife.79743.sa1
Author response https://doi.org/10.7554/eLife.79743.sa2

# Additional files

## Supplementary files

• Supplementary file 1. Mathematical model parameters.

• Supplementary file 2. Primers used in this study.

• Supplementary file 3. smFISH probe sequences used to detect FLC Venus transcripts. These probes were labeled with Quasar570.

• Supplementary file 4. smFISH probe sequences from *Duncan et al., 2016* (**A**) used to detect unspliced PP2A transcripts and (**B**) used to detect *FLC* sense spliced. Both probe sets were labeled with Quasar670.

• Supplementary file 5. Raw data from this study.

• MDAR checklist

## Data availability

All microscopy data has been deposited to the BioImage Archive with Accession code S-BIAD425 (https://www.ebi.ac.uk/biostudies/studies/S-BIAD425). Derived quantification data from images, as well as qPCR data, are provided as *Supplementary file 5*. All code is available through GitHub repositories, as described in the 'Materials and methods'.

The following dataset was generated:

| Author(s) | Year | Dataset title | Dataset URL | Database and Identifier |
|---|---|---|---|---|
| Antoniou-Kourounioti R, Meschischi A, Reeck S, Berry S, Menon G, Zhao Y, Fozard JA, Holmes T, Wang H, Hartley M, Dean C, Rosa S, Howard M | 2022 | Integrating analog and digital modes of gene expression at Arabidopsis FLC | https://www.ebi.ac.uk/biostudies/BioImages/studies/S-BIAD425?query=S-BIAD425 | BioImage Archive, S-BIAD425 |

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
