## [Editor Report]

Regulation of gene expression in many biological systems occurs either digitally where gene expression is either on or off or through an analog mode with graded modulation of gene expression. In this study, the authors report how these two regulatory modes are integrated into a one-way switch pattern to control the expression of the *Arabidopsis* floral repressor gene *FLOWERING LOCUS C* (*FLC*). The results of their work lead the authors to propose that analog regulation in the autonomous flowering pathway precedes digital regulation conferred by Polycomb silencing before cold exposure, and that this temporal switch correlates with the strength of transcription at the *FLC* locus in different genetic backgrounds.

---

## [Decision Letter]

**Decision letter after peer review:**

Thank you for submitting your article "Integrating analog and digital modes of gene expression at Arabidopsis FLC" for consideration by *eLife*. Your article has been reviewed by 3 peer reviewers, and the evaluation has been overseen by the Reviewing Editor, Hao Yu, and Naama Barkai as the Senior Editor. The following individual involved in review of your submission has agreed to reveal their identity: Teva Vernoux (Reviewer #1).

The reviewers have discussed their reviews with one another. While there was a common interest about the topic and the relevant findings, some concerns on the methodologies, genetic materials, and interpretation of results as listed below in the section of "Essential revisions" need to be addressed before the manuscript could be further considered.

Essential revisions:

1. The authors should consider using live-imaging to track the cell-level correlation between ON/OFF states of FLC and FLC expression levels as well as their link with cell division. These results are required to support the corresponding correlations established by the results obtained at the cell population level.

2. The nature of the genetic materials (e.g. fca-3 and FLC-Venus) used or created in this study should be clearly characterized and described in the manuscript. Notably, the intermediate mutant fca-3 exhibits comparable FCA expression levels to the strong mutant fca-1 (Figure 3B). How the mutations in these alleles eventually cause different FLC expression levels and flowering time could be relevant to different types of FCA forms produced in these mutants. As fca-1 and fca-3 are crucial genetic material used in this study, more detailed information on these two mutants and rationales of selecting fca-3 as the intermediate mutant should be provided for interpreting the results obtained in this study.

3. The authors utilized the root system with transgenes in mutant backgrounds to observe and model the repression of FLC expression. In some experiments, the authors compared the results obtained from different tissues (e.g. qPCR measurement of FLC in whole plants in Figure 3A versus FLC signal in root cells in other panels). As FLC mainly functions in leaves and the shoot apical meristem (SAM) to regulate flowering time, how the molecular logic deduced from root cells could be applicable to FLC roles in leaves and SAM should be justified with concrete evidence.

4. For the time-course experiments (e.g. Figure 3A and 4B), examination of FLC expression or active copies beyond the floral transition stage (particularly for Ler) could be biologically irrelevant.

5. The thresholding definition of digitally ON and OFF of FLC expression seems too arbitrary. This needs to be better justified, and, if necessary, redefined.

6. The authors should consider whether it is more appropriate to use medians rather than means for the plots exhibiting skewed distributions. In addition, biological repeats and statistical analyses are required for most of the quantitative data in this manuscript.

7. Although this study proposes a new paradigm for gradual repression of FLC by the combined analog and digital modes, how this model is applicable to plants in different genetic backgrounds and integrated with other regulatory mechanisms pertaining to FLC regulation should be discussed.

Please also take into consideration the other specific comments from the reviewers below to revise the manuscript.

*Reviewer #1 (Recommendations for the authors):*

1. To go beyond cell population-level correlations, the authors should use live-imaging to track cells and show that their probability to switch FLC off is indeed correlated to the level of FLC expression. This cell-level correlation would help to build a much stronger case for the importance of analog regulation in determining the dynamics of FLC. It is also necessary to claim that analog regulation comes before digital regulation as the authors do, otherwise this claim is not really supported by the data.

2. Figure 2D (v): to claim that there is a heritable component in the ON/OFF state the authors need to show more than one image.

3. About the half-lives of FLC mRNA and FLC protein: the measurements effectively indicate relatively short half-lives but could the authors show further evidence (e.g. using a simple model) that these values are compatible with the interpretation of their results. This would be more convincing.

4. As the link between division and digital regulation is important to the model, couldn't the authors use live-imaging to show this functional link? That would be an important addition to the manuscript.

5. Concerning the model again, the authors could show that they reproduce also the wild-type and fca-1 situations. Notably for the wild-type, the mRNA and protein distribution are relatively similar despite the clear quantitative difference and this would strengthen the modeling analysis.

6. How do the authors explain that fca-1 still flowers despite the absence of switch? Does the s.itch occur later? Is this an artifact resulting from the use of roots instead of shoots?

7. The authors could show that the digital regulation of FLC is also dependent on Polycomb in absence of cold treatment (or cite existing work if this evidence already exists). This would help to better connect this work to what is known in the case of cold exposure.

8. It would be important that the authors discuss more the relevance of their findings to what occurs in wild-type background and to better highlight why their findings are important for understanding how flowering is controlled. This is missing at the moment. Is it reasonable to speculate that FRI allelic series could represent a situation somehow equivalent to what has been studied in this work? In relation to this, I am not convinced that the authors can claim that analog regulation is dependent on the autonomous pathway. As such their data could rather indicate that the autonomous pathway in Ler prevent analog regulation, allowing for an efficient digital regulation of FLC. If an analog regulation of FLC indeed exists in the wild, one could envision that it rather arises from an equilibrium between at least the FRI and autonomous pathways. The authors should revise this claim and discuss how they envision that analog regulation could occur in wild-type plants.

*Reviewer #2 (Recommendations for the authors):*

The nature of the fca-3 mutant should be described in detail.

What is the phenotype (flowering time) of FLC-venus in fca-1 and fca-3? In addition, how many independent lines were used? Do they behave similarly?

Whether what the authors observe is "biologically" relevant is a critical point the authors should carefully address.

The authors should measure the functional form of FCA transcripts in Figure 3B, and as discussed earlier, the description of fca-1 and fca-3 alleles would be helpful for readers.

*Reviewer #3 (Recommendations for the authors):*

When assessing the differences between the FLC transcript and protein levels in the histograms for Ler, fca-3, and fca-1, one can see a much larger gap between fca-3 and fca-1 levels than fca-3 and Ler. Could the authors better justify the use of fca-3 as an 'intermediate' state to assess the interplay between digital and analog mode? Given the outcome of the quantifications, perhaps the word 'intermediate' might need to be reconsidered. Adding more comments about these levels during the text (better describe its medians rather than means) would be also appreciated.

The model needs some further clarity to better connect it with the data, and, if possible, it would be ideal if this could be used for understanding the different regulatory modes.

I would suggest the authors use the model to provide examples such the different regulatory modes (digital, analog and combined digital-analog regulation) can be recapitulated and more easily connected to the data. Yet, perhaps this is not the purpose of that model and this connection is not that easy, given how the model itself has been formulated. If this is the case, I invite the authors to consider reformulating the model to clarify the different regulatory modes and better connect it with the data.

In line 501 it says there is a "manual fitting" process. Could you explain that more formally? This seems arbitrary and I am wondering why a curve or distribution fitting algorithm couldn't work better.

I find the idea of connecting the simulated cell states with the FLC-Venus experimental data quite original (as in Lines 505-508). However, I am a bit confused as to why the authors have selected the equations in 505-507. Why is this the case? And are there other distributions that may work better? (say a Poisson distribution, as that may give you a better indication of dividing since you are counting the number of cycles a cell goes through).

From the time course in Figure 3 it is concluded that there is 'one-way switching to an OFF' state in fca-3. I was wondering whether the authors could discuss further the possibility of switching from 'OFF to ON'. Presumably, by visual observation of the experimental data one could think this may happen – e.g. when looking at a given time point in cells along the same row in some cases in fca-3 and fca-1. Furthermore, could the model help in evaluating how things would change if OFF-ON transitions are included?

In Figure 3A, the authors perform qPCR in the whole plants while the rest of the figure focuses on looking at silencing at the root tips. For consistency, wouldn't it be possible to perform the qPCR in root tips? I am aware this would be a significant amount of work, but the current conclusions extracted from Figure 3A are not clear to me, nor whether this qPCR quantification in the whole plant helps to find digital silencing that is FCA independent. In terms of conclusions – please revise the used (type of) tests; it is strange to me that fca-1 does not show a significant decrease, as commented in lines 221 and 233.

Could the authors comment on the bimodal distributions potentially observed in fca-1?

Would it be possible to quantify the dilution effects from the experimental data? I was wondering whether this could be a possible explanation for the decrease in FLC-VENUS levels in the time course, especially for Ler and fca-3, as an alternative to digital silencing.

Figure 2S3A – One can see the sphere to quantify the fluorescence intensity in some cases is partly outside of the measured cell (especially when the cell is smaller in width than the diameter of the sphere), indicating this computation will dilute the signal. This can drive to potentially artifacts and negative concentrations. Could the authors correct this effect?

Histogram Figures, such as Figure 2E, have negative concentrations, so this should be revised. Perhaps it is related to a previous point on the outlined spheres to quantify the fluorescence intensity.

[Editors' note: further revisions were suggested prior to acceptance, as described below.]

Thank you for resubmitting your work entitled "Integrating analog and digital modes of gene expression at Arabidopsis *FLC*" for further consideration by *eLife*. Your revised article has been evaluated by Detlef Weigel (Senior Editor) and a Reviewing Editor.

The manuscript has been improved but there are some remaining issues that need to be addressed, as outlined below:

1. Why different analog expression of FLC occurs in different fca alleles should be discussed.

2. It will be helpful to include an internal control in the FLC-venus experiments to compare fluorescence levels in ON cells with varying degrees of FLC-venus expression.

3. Clarifications on some major issues raised by reviewers, such as in-depth quantification in non-root tissues and actual meaning of the integration of digital and analog regulation, should be briefly incorporated into the manuscript.

Please also take into consideration the other specific comments from the reviewers below to revise the manuscript.

*Reviewer #1 (Recommendations for the authors):*

In this revised version, the authors have consolidated their data and their text, including the discussion. One can certainly agree, that given the difficulty to obtain reliable live-imaging data, their population-level analysis from imaging snapshots provides a sufficiently solid basis to the idea of a combination of an analog and digital regulation of the expression of FLC. Demonstrating expression of FLC in shoot tissues is also certainly strengthening the hypothesis that the mode of action seen in the root is relevant to shoot tissues. However, it would be nice if the authors could mention at the end of the discussion that, finding a way to address these two questions in the future will be important to explore further how the regulation they have identified acts in the regulation of flowering.

*Reviewer #2 (Recommendations for the authors):*

I appreciate that the authors have addressed several criticisms from reviewers in this revision.

This study reports the existence of an "analog" mode of regulation in addition to the previously reported "ON-OFF" switch for FLC regulation. By comparing the levels of FLC mRNA/protein in different genetic backgrounds, the authors provide evidence for the analog mode of FLC regulation. Although time-course experiments suggest that the "ON-OFF" switch occurs in ON cells regardless of FLC levels, the mechanism behind the analog difference in FLC levels in fca-1 and fca-3 mutants remains unclear.

I also understand that the authors empirically selected fca-1 and fca-3 (and fca-4 in this revision) mutants based on distinct levels of FLC (and flowering time). The main conclusion of the manuscript hinges on the different behaviors of FLC (analog) observed in these genetic backgrounds. Therefore, it is crucial to address how the various alleles of fca influence FLC regulation. However, the study did not address why different alleles of fca resulted in different analog mode of FLC expression.

As a result, the study is primarily descriptive.

1. The study would benefit from an internal control in the FLC-venus experiments to compare fluorescence levels in ON cells with varying degrees of FLC-venus expression (after all, the authors are comparing fluorescence in different cells among different transgenic lines in different genetic backgrounds).

2. I appreciate the detailed description of different alleles of fca used in this revision. However, I am still puzzled how fca-1 and fca-3, which both failed to produce "functional" FCA protein, resulted in different analog expressions of FLC. The authors discussed the analog regulation of FLC as an early regulation of FLC and the level of transcription affects the Polycomb switch. It would be more relevant to discuss how the different levels of FLC transcription are established in fca-1 and fca-3 mutants.

*Reviewer #3 (Recommendations for the authors):*

I would like to thank first the authors for all the work performed on the previous version and the addressed comments. I find the manuscript has improved, although I still have several issues I would like to comment on:

1. I appreciate the effort to study the FLC behavior in leaves. Yet, by eye it is difficult to distinguish whether Ler and fca-3 are exhibiting different behaviors, as the authors claim it is occurring in the root (Ler and fca-3 seem to express at the same levels in Figure 2S4). Hence, despite of the challenge, I believe quantification would be important to support the claims by the authors, although I am aware the authors were told by the editor this could be done 'without the need of in-depth quantification'. Otherwise, could the authors comment on that, and clarify in the manuscript?

2. In my understanding, the ON transcriptional state between Ler and fca-3 is having the same (median) levels, according to Figure 2S2, which tells us that the differences between these two lines is just in the digital mode, and not the analog mode. Hence, if I understand well, digital and analog mode is not seen (or I could not clearly see) in the same line at the same time – at least, at the transcriptional level. If this is the case, I think the claim of the integration of digital and analog might need to be revised, given the current data is not showing both regulations in the same line. If this is not the case, I believe I am misunderstanding the interpretation of your data, and further clarifications would be needed.

3. I agreed with the comment 3 by Reviewer 1, and the authors thought it is not needed to provide further evidence, given the differences between the half-lives timescales (hours) with the developmental timescales (weeks). Yet, I am wondering whether the timescale to compare to understand in part the resulting distributions should also be the division timescale (a day). Given this manuscript relies on quantitative evidences, I believe it is still unclear for the reader whether dilution effects or the protein or RNA half-lives (perhaps together with noise) might broaden the histograms, conferring the false impression of analog regulation. Could slightly different dilution (and noisy) rates between the different lines account for the different protein levels shown in Figure 2? I think given the central scope of this manuscript on distinguishing what is analog and what is digital, this point needs more justification and/or clarification (ideally, a model could help).

4. I understand that the authors might find the manual fitting good enough for them, but they derive conclusions out of it, in particular now they give numbers about the OFF-ON rates in relation to the ON-OFF rates. I think that given the quantitative scope of this work, it would be more ideal to have a more reproducible fitting to have solid conclusions.

5. Overall, although I appreciate the very good and interesting work from the authors, I still find there are aspects of clarity of the manuscript that need to be revised, that better link the data from the conclusions extracted by the authors. For instance, as I said above it is not clear to me what the authors really mean by 'the integration of the digital and analog regulation', given I do not clearly see such two modes acting in the same line, and I am still hesitant about not seeing a bimodal distribution whose expression levels could be modulated, which would be the major evidence to me. Also, lines in the abstract 39-40 are not totally clear to me; I understand there is a slow time scale for the digital silencing, but I am not totally sure about associating a timescale to the analog regulation, and therefore, to compare both timescales. (Maintenance of expression levels does not necessarily show to me that analog precedes the digital silencing). I would appreciate more clarifications in this directions.

6. I noticed that the manuscript version with tracked changes (which is the version I initially read in detail) is not the same as the non-tracked version (e.g. see lines 553 and 596, which refer to Sup File 3 and Sup. File 1, and compare it with the tracked version, in which the equivalent sentences would refer to Sup. File 1 and Supp Table 4, respectively). I hope there are not further major differences between the tracked and non-tracked manuscripts.

---

## [Author Response]

Essential revisions:1. The authors should consider using live-imaging to track the cell-level correlation between ON/OFF states of FLC and FLC expression levels as well as their link with cell division. These results are required to support the corresponding correlations established by the results obtained at the cell population level.

We have put considerable effort into live imagery, as suggested by the reviewers, in both *fca-3* and also a new background *fca-4*, which is a different mutation, but which otherwise behaves similarly to *fca-3* (Figure 2S3). We believe the similar behaviour of both intermediate *fca* lines adds further robustness to our study. However, despite some early success, we have not been able to reliably generate dynamical timecourses. The reason stems from the very low levels of FLC-Venus in *fca3/fca-4*, and corresponding high laser power required for imaging. Under such conditions, the roots are clearly unhappy, cells only elongate and stop dividing, presumably due to cell cycle arrest, and we also cannot observe any switching. We have tried multiple methodologies, including on plates, chambered coverslips (with roots growing between the coverglass and a block of agar), and an agarblock chamber containing perfluorodecalin to maintain constant levels of oxygen while imaging, but without success. Very rarely (2 roots only, 1 in *fca-3* and 1 in *fca-4*), we found roots that did still grow, with cells dividing, and in both cases we were able to identify switching from ON to OFF. However, we do not think this is sufficiently reproducible to be included in the manuscript. Instead, analysing high-precision population-level FLC data from imaging snapshots, as we have already presented in the manuscript, is the only feasible approach. We emphasise that this approach is fully sufficient to justify our conclusions: the reducing fraction of cells with visible *FLC* mRNA/protein can only be achieved through individual cells predominantly switching off *FLC* expression over time.

These new results do show, however, that an active cell cycle is required for switching, but we do not have evidence that this is specifically connected to division. We have also examined this in our computational model and found that our results are unchanged if we drop this connection. Therefore, we have simplified the model to remove the cell-division link to switching, so that a switch from ON to OFF can happen at any time, independently of the cell-cycle stage, with a constant probability per unit time, provided that the cell has an active cell cycle (which all the cells we simulate do, as they are all in the division zone of the root). A switch from OFF-to-ON with a low 7% rate relative to the ON-to-OFF transition was also added to this model (see response to Reviewer 3’s comments below).

2. The nature of the genetic materials (e.g. fca-3 and FLC-Venus) used or created in this study should be clearly characterized and described in the manuscript. Notably, the intermediate mutant fca-3 exhibits comparable FCA expression levels to the strong mutant fca-1 (Figure 3B). How the mutations in these alleles eventually cause different FLC expression levels and flowering time could be relevant to different types of FCA forms produced in these mutants. As fca-1 and fca-3 are crucial genetic material used in this study, more detailed information on these two mutants and rationales of selecting fca-3 as the intermediate mutant should be provided for interpreting the results obtained in this study.

We thank the reviewers for pointing out this omission. We have added much more information on the genotypes in the methods of the manuscript. We emphasise, however, that the rationale for selecting *fca-3* as an intermediate mutant was empirical: namely, it generates an intermediate level of *FLC* expression (Figure 1C and Figure 1S1).

3. The authors utilized the root system with transgenes in mutant backgrounds to observe and model the repression of FLC expression. In some experiments, the authors compared the results obtained from different tissues (e.g. qPCR measurement of FLC in whole plants in Figure 3A versus FLC signal in root cells in other panels). As FLC mainly functions in leaves and the shoot apical meristem (SAM) to regulate flowering time, how the molecular logic deduced from root cells could be applicable to FLC roles in leaves and SAM should be justified with concrete evidence.

The developmental structure, with clonal cell files, and transparency of the Arabidopsis root make it an ideal tissue for our imaging analysis. Imaging green tissues with very low expression, such as FLCVenus in *fca-3* mutant, is extremely challenging (see point 1 above). However, we have now managed to obtain images of FLC-Venus in young leaf tissues using ClearSee (a method that diminishes autofluorescence while preserving fluorescent protein signal; see Kurihara et al., ClearSee: a rapid optical clearing reagent for whole-plant fluorescence imaging, Development (2015)). Using this method, we have now imaged FLC in young leaves (Figure 2S4) in Ler, *fca-3* and *fca1*. As can be seen from this image, we observe a similar pattern as in the roots: (Ler cells OFF, *fca-1* cells ON, *fca-3* cells some OFF and some cells ON, with ON cells at an analog reduced intensity compared to *fca-1*). In the non-root tissue, even for imaging snapshots, full statistics would require a completely new image analysis pipeline for young leaves rather than roots. Developing such a pipeline for these tissues is, we feel, beyond the scope of this work, and would also require huge amounts of additional image acquisition to gather enough cells. Instead, we show example images in the revised manuscript to make it visually clear that *FLC* in young leaves does indeed behave similarly to the roots.

4. For the time-course experiments (e.g. Figure 3A and 4B), examination of FLC expression or active copies beyond the floral transition stage (particularly for Ler) could be biologically irrelevant.

Indeed, Ler is the only line that has transitioned to flowering during the experiment, with both *fca* lines being late flowering mutants. We totally agree that for Ler, later timepoints may be biologically irrelevant. It is used in this case as a negative control for the imaging, since *FLC* in Ler was already mostly OFF from the first timepoint and no biological conclusions are drawn from the later times. We have added a comment to this effect in the Results section, also clarifying in the discussion that our focus is on the early regulation of *FLC*. Therefore, by looking at the young seedling in wildtype Ler, as we and others have previously, we are already looking too late to capture the switching of *FLC* to OFF. However, we expect that this combination of analog and digital regulation will be highly relevant to *FLC* regulation in wild-type plants in different accessions, partly leading to the differences in autumn *FLC* levels that were shown to be so important in the wild (Hepworth et al. 2020). We have added more discussion on this point also in relation to comment 7 below.

5. The thresholding definition of digitally ON and OFF of FLC expression seems too arbitrary. This needs to be better justified, and, if necessary, redefined.

We agree with the reviewer that the choice of threshold is somewhat arbitrary, as the ON and OFF distributions are overlapping. Specifically for the Venus fluorescence intensity data, we have now removed the statements in the manuscript that were related to quantification based on these thresholds, and avoided using thresholds where possible. As we now make clearer in the manuscript, we now use a threshold to compare only the tails of the fluorescence intensity *fca-3* distributions at the three timepoints (Figure 3S4). We use the high threshold of 1, to look only at the tail of the ON cells.

We have retained thresholds from smFISH experiments, where measurements of mRNAs are absolute. In this case, we tested other thresholds and our conclusions are robust to the choice of threshold, as shown in new Figure 2S2. We have, however, employed a slightly higher threshold (more than 3 visible foci rather than 1 for ON cells), as we believe this accounts better for spurious noisy foci present at very low levels.

6. The authors should consider whether it is more appropriate to use medians rather than means for the plots exhibiting skewed distributions. In addition, biological repeats and statistical analyses are required for most of the quantitative data in this manuscript.

We agree that showing medians may be more appropriate and we have now added the medians as well as the means to all relevant plots where we make comparisons in the manuscript, also allowing a comparison for the reader. The statistical test used to compare distributions is the non-parametric Kruskal–Wallis test, so though we were showing the means, this was not what was being compared in the statistical tests. We have now also added statistical tests for Figure 1C, 3B which were previously missing. We performed biological repeats and data shown is combined cells from two or three biological replicates. The only exception was where the tissue was labelled as cortex/epidermis, where data from only one experiment were used. Therefore, we have now removed this panel. We have also now added number of experiment repeats in the figure legends. Furthermore, the supplementary file indicates from which replicate each cell measurement originated.

7. Although this study proposes a new paradigm for gradual repression of FLC by the combined analog and digital modes, how this model is applicable to plants in different genetic backgrounds and integrated with other regulatory mechanisms pertaining to FLC regulation should be discussed.

We have now added discussion on this excellent point into the Discussion section.

Please also take into consideration the other specific comments from the reviewers below to revise the manuscript.Reviewer #1 (Recommendations for the authors):1. To go beyond cell population-level correlations, the authors should use live-imaging to track cells and show that their probability to switch FLC off is indeed correlated to the level of FLC expression. This cell-level correlation would help to build a much stronger case for the importance of analog regulation in determining the dynamics of FLC. It is also necessary to claim that analog regulation comes before digital regulation as the authors do, otherwise this claim is not really supported by the data.

See response to Essential comment 1. Switching of cells from an analog into a digitally silenced state is, we believe, still clear from the population level data: comparing ON cells in *fca-3* versus *fca-1* at the 7-day timepoint, it is clear that the former has lower FLC levels (both protein and RNA), thus demonstrating analog regulation. Over time, at a population level in *fca-3*, the proportion of ON cells decreases, with the fraction of cells with no visible FLC increasing, thus supporting a predominantly one-way switch into a digitally silenced state. Our modelling also supports a predominantly one-way switch to digital OFF, although a much lower rate of switching back to analog ON is also possible, as we now show. Overall, we find that analog regulation does predominantly precede digital: if this were not so, there would be no progressive reduction in the number of ON cells.

2. Figure 2D (v): to claim that there is a heritable component in the ON/OFF state the authors need to show more than one image.

We have now added yellow boxes in Figure 2S3 to show additional examples of short files of ON cells in *fca-3* and *fca-4.* To further improve the interpretation of this image (and all others in the manuscript) we have changed the presentation of the imaging using a different colourmap to enhance clarity.

3. About the half-lives of FLC mRNA and FLC protein: the measurements effectively indicate relatively short half-lives but could the authors show further evidence (e.g. using a simple model) that these values are compatible with the interpretation of their results. This would be more convincing.

The measured half-lives are indeed short (hours) compared to the developmental timescale (weeks). For this reason, the decay of protein/RNA can't cause the analog reduction we see in *fca-3*. In a scenario where all the cells in *fca-3* switch off early on, then all the protein/RNA would be lost within a day, which is not what we observe. This rapid decay, with all protein/RNA lost within a day of switching, is also why we don't incorporate these dynamics into the model: there will be a short period of intermediate FLC levels immediately after a cell switches OFF, but this is a very minor effect given the weeks-long duration of development in our experiments.

4. As the link between division and digital regulation is important to the model, couldn't the authors use live-imaging to show this functional link? That would be an important addition to the manuscript.

See response to Essential comment 1.

5. Concerning the model again, the authors could show that they reproduce also the wild-type and fca-1 situations. Notably for the wild-type, the mRNA and protein distribution are relatively similar despite the clear quantitative difference and this would strengthen the modeling analysis.

For the wild-type and *fca-1* situations there is no switching in the model, and hence no dynamical changes in the FLC protein levels. As the FLC levels in the ON or OFF states are simply fit to the data using log-normal distributions, this would simply be a fitting exercise for *fca-1* and Ler, and little would be learnt. Hence, we have not pursued this line of analysis.

6. How do the authors explain that fca-1 still flowers despite the absence of switch? Does the s.itch occur later? Is this an artifact resulting from the use of roots instead of shoots?

Highly expressing *FLC* lines and mutants, such as Col*FRI* and *fca-9*, often used for vernalization studies, are late flowering, but do eventually flower even with no decrease in *FLC* levels (and so no switching). This is not an artifact of using roots versus shoots, and presumably arises from there being multiple inputs into the flowering decision which can allow the FLC-mediated flowering inhibition to eventually be overcome.

7. The authors could show that the digital regulation of FLC is also dependent on Polycomb in absence of cold treatment (or cite existing work if this evidence already exists). This would help to better connect this work to what is known in the case of cold exposure.

H3K27me3, the Polycomb mark, is present at *FLC* in wildtype Ler and Col-0, where *FLC* is silenced in the absence of cold treatment. In Col*FRI* and the natural variant Lov-1, both of which have high *FLC* in the absence of cold treatment, H3K27me3 levels at *FLC* are low. In the *clf* mutant in Col-0 it has been observed that *FLC* is upregulated and H3K27me3 is reduced (Shu et al., 2019). This evidence does not prove that the digitally-silenced state in this case is Polycomb dependent, but we propose it as a likely explanation considering the analogy with the cold-dependent digital state. We now say this explicitly in the Discussion.

8. It would be important that the authors discuss more the relevance of their findings to what occurs in wild-type background and to better highlight why their findings are important for understanding how flowering is controlled. This is missing at the moment. Is it reasonable to speculate that FRI allelic series could represent a situation somehow equivalent to what has been studied in this work? In relation to this, I am not convinced that the authors can claim that analog regulation is dependent on the autonomous pathway. As such their data could rather indicate that the autonomous pathway in Ler prevent analog regulation, allowing for an efficient digital regulation of FLC. If an analog regulation of FLC indeed exists in the wild, one could envision that it rather arises from an equilibrium between at least the FRI and autonomous pathways. The authors should revise this claim and discuss how they envision that analog regulation could occur in wild-type plants.

We agree with the reviewer’s comment that analog regulation most likely arises from an equilibrium between FRI and autonomous pathways and have added to the manuscript that we would expect a FRI allelic series to be equivalent. Additional discussion on the importance of these findings for flowering regulation in natural variants has also now been added. Regarding the reviewer’s comment that the autonomous pathway prevents analog regulation in Ler, we would in principle agree, but are using a slightly different framing of what we believe is effectively the same statement, which is that the autonomous pathway in Ler prevents analog upregulation, giving low analog levels that are easily overturned by Polycomb.

Reviewer #2 (Recommendations for the authors):The nature of the fca-3 mutant should be described in detail.What is the phenotype (flowering time) of FLC-venus in fca-1 and fca-3? In addition, how many independent lines were used? Do they behave similarly?

It was observed that with the additional *FLC* gene (in the form of the *FLC-Venus*), flowering is delayed as expected. However, this was not quantified in this work. Instead, we validated that the expression of the transgene was equivalent to endogeneous between genotypes, as shown in Figure 1S1, supporting that this is an appropriate readout for *FLC* expression. One line for each genotype was selected and used in this work. In addition, we also now use *fca-4*, which has similar expression to *fca-3*, and where FLC-Venus also behaves similarly to the *fca-3* case (Figure 1S1, 2S3).

Whether what the authors observe is "biologically" relevant is a critical point the authors should carefully address.The authors should measure the functional form of FCA transcripts in Figure 3B, and as discussed earlier, the description of fca-1 and fca-3 alleles would be helpful for readers.

See response to Essential comment 2.

Reviewer #3 (Recommendations for the authors):When assessing the differences between the FLC transcript and protein levels in the histograms for Ler, fca-3, and fca-1, one can see a much larger gap between fca-3 and fca-1 levels than fca-3 and Ler. Could the authors better justify the use of fca-3 as an 'intermediate' state to assess the interplay between digital and analog mode? Given the outcome of the quantifications, perhaps the word 'intermediate' might need to be reconsidered. Adding more comments about these levels during the text (better describe its medians rather than means) would be also appreciated.

As described elsewhere, we now also use the medians as well as the means to assess the FLC levels. It is true that the FLC levels in *fca-3* are much closer to Ler than to *fca-1*. Nevertheless, they are clearly distinctly higher than in Ler. For this reason, we think the description “intermediate” (whose definition means coming between two things) is appropriate.

The model needs some further clarity to better connect it with the data, and, if possible, it would be ideal if this could be used for understanding the different regulatory modes.I would suggest the authors use the model to provide examples such the different regulatory modes (digital, analog and combined digital-analog regulation) can be recapitulated and more easily connected to the data. Yet, perhaps this is not the purpose of that model and this connection is not that easy, given how the model itself has been formulated. If this is the case, I invite the authors to consider reformulating the model to clarify the different regulatory modes and better connect it with the data.

As the reviewer points out, reworking the model and presentation in the way suggested is not straightforward. While it is more conceptually satisfying, it tends to obscure the connection with our data, as this connection becomes only one out of several possible regulatory modes. Instead, we have expanded the Discussion to cover the possible range of regulatory modes at *FLC*, mentioning experimental cases (known and hypothetical) where only analog (no Polycomb) or only digital (cold) regulation might occur.

In line 501 it says there is a "manual fitting" process. Could you explain that more formally? This seems arbitrary and I am wondering why a curve or distribution fitting algorithm couldn't work better.

In the manual fitting, the parameters were adjusted by hand until the model fit was judged to be satisfactory. No algorithms were used, as this was found to be unnecessary to generate fits that were close to the experimental data. Potentially the fits could be made even better by using more sophisticated algorithms, but given the simplicity of the underlying model we think the current manual fits are already quite satisfactory.

I find the idea of connecting the simulated cell states with the FLC-Venus experimental data quite original (as in Lines 505-508). However, I am a bit confused as to why the authors have selected the equations in 505-507. Why is this the case? And are there other distributions that may work better? (say a Poisson distribution, as that may give you a better indication of dividing since you are counting the number of cycles a cell goes through).

We have added more clarification to justify the choice of the equations in 505-507, including the choice of the log-normal distribution. We note that division and switching is not included in those equations, which define the distribution of FLC-Venus signal given the ON/OFF state of the *FLC* copies in the cells.

From the time course in Figure 3 it is concluded that there is 'one-way switching to an OFF' state in fca-3. I was wondering whether the authors could discuss further the possibility of switching from 'OFF to ON'. Presumably, by visual observation of the experimental data one could think this may happen – e.g. when looking at a given time point in cells along the same row in some cases in fca-3 and fca-1. Furthermore, could the model help in evaluating how things would change if OFF-ON transitions are included?

We have now included switching from OFF to ON in the model, thus weakening our assumptions. We found that this is possible as long as the switch ON rate is much lower than the switch OFF rate (to allow for the gradual reduction in ON cells observed in the time-course).

In Figure 3A, the authors perform qPCR in the whole plants while the rest of the figure focuses on looking at silencing at the root tips. For consistency, wouldn't it be possible to perform the qPCR in root tips? I am aware this would be a significant amount of work, but the current conclusions extracted from Figure 3A are not clear to me, nor whether this qPCR quantification in the whole plant helps to find digital silencing that is FCA independent. In terms of conclusions – please revise the used (type of) tests; it is strange to me that fca-1 does not show a significant decrease, as commented in lines 221 and 233.

qPCR at root tips is extremely difficult as it requires a large number plants and laborious methodology. Instead, we have taken the opposite approach which is to retain whole plant qPCR, but use single cell imagery in other parts of the plant (the young leaves). This reinforces our conclusion that the reduction in expression found at the whole plant level in *fca-3* in Figure 3A is due to more and more cells switching to a digitally silenced state over time. Finally, we do not think it appropriate to revise the statistical tests, which are the accepted methodology. We have doublechecked the result of the statistical test for *fca-1* in Figure 3A and it is indeed not significant. We believe this is because of the wide error bars at the first time point. However, a separate experiment in non-transgenic plants showed similar trends but in that case the decrease was significant. We have included this data in the manuscript as Figure 3S2.

Could the authors comment on the bimodal distributions potentially observed in fca-1?

The evidence for bimodality in *fca-1* is not strong and it is possible that this is just a statistical fluctuation. We therefore feel it would be unhelpful to speculate further.

Would it be possible to quantify the dilution effects from the experimental data? I was wondering whether this could be a possible explanation for the decrease in FLC-VENUS levels in the time course, especially for Ler and fca-3, as an alternative to digital silencing.

For the distributions of cells we image over our time-course, their size does not alter over time, rather the number of high intensity cells changes. Therefore dilution seems unlikely to be the underlying explanation for the decreasing trend.

Figure 2S3A – One can see the sphere to quantify the fluorescence intensity in some cases is partly outside of the measured cell (especially when the cell is smaller in width than the diameter of the sphere), indicating this computation will dilute the signal. This can drive to potentially artifacts and negative concentrations. Could the authors correct this effect?

This effect had been corrected in the calculations, but this was unfortunately not fully described in the manuscript. We have now corrected the description in methods section “FLC-Venus fluorescence intensity” to make this clear.

Histogram Figures, such as Figure 2E, have negative concentrations, so this should be revised. Perhaps it is related to a previous point on the outlined spheres to quantify the fluorescence intensity.

The negative values appear because the cell background intensity is subtracted from the nucleus intensity, and for OFF cells the two are the same except for noise which can be positive or negative. This is now explained in the methods.

[Editors' note: further revisions were suggested prior to acceptance, as described below.]

The manuscript has been improved but there are some remaining issues that need to be addressed, as outlined below:1. Why different analog expression of FLC occurs in different fca alleles should be discussed.

We do not fully understand the exact mechanism by which the analog regulation is achieved. However, we do know that in *fca-3* a compromised/truncated version of the protein is being produced and this leads to a less efficient downregulation of *FLC*. We have now added detail of the differences between the genotypes in the main text in the Introduction.

2. It will be helpful to include an internal control in the FLC-venus experiments to compare fluorescence levels in ON cells with varying degrees of FLC-venus expression.

The FLC-Venus transgenic lines analysed in our study were generated by crossing, not transformation, and are therefore not independent transgenic lines. We now explain this directly where we discuss these issues in the main text. Therefore, any differences in FLC-Venus level do not reflect differences associated with the insertion site but differences in the genetic background (in this case, different *fca* alleles). Moreover, the different genotypes were always imaged together while using the same settings, which is a common practice when comparing fluorescent protein levels. Furthermore, the differences between Ler, *fca-1* and *fca-3* are very large and, for *fca-1* and *fca-3*, are supported by RNA FISH experiments that rely on counts of single mRNA molecules and correspond to absolute numbers, thereby eliminating any need for internal controls. For these reasons we do not think that such an internal control is necessary. Furthermore, including such a control would require crossing with a marker line and re-selection of FLC-Venus homozygous lines. This would therefore require ~12 months of work for the generation of these lines plus redoing all imaging experiments and data analysis.

3. Clarifications on some major issues raised by reviewers, such as in-depth quantification in non-root tissues and actual meaning of the integration of digital and analog regulation, should be briefly incorporated into the manuscript.

For “in-depth quantification in non-root tissues”, a new figure panel has been added with the quantification of non-root tissues, with a statistical analysis described in the legend. For the “meaning of the integration of digital and analog regulation”, the Discussion has been edited to streamline and more directly explain this aspect.

Reviewer #1 (Recommendations for the authors):In this revised version, the authors have consolidated their data and their text, including the discussion. One can certainly agree, that given the difficulty to obtain reliable live-imaging data, their population-level analysis from imaging snapshots provides a sufficiently solid basis to the idea of a combination of an analog and digital regulation of the expression of FLC. Demonstrating expression of FLC in shoot tissues is also certainly strengthening the hypothesis that the mode of action seen in the root is relevant to shoot tissues. However, it would be nice if the authors could mention at the end of the discussion that, finding a way to address these two questions in the future will be important to explore further how the regulation they have identified acts in the regulation of flowering.

This aspect has now been added in the Discussion.

Reviewer #2 (Recommendations for the authors):I appreciate that the authors have addressed several criticisms from reviewers in this revision.This study reports the existence of an "analog" mode of regulation in addition to the previously reported "ON-OFF" switch for FLC regulation. By comparing the levels of FLC mRNA/protein in different genetic backgrounds, the authors provide evidence for the analog mode of FLC regulation. Although time-course experiments suggest that the "ON-OFF" switch occurs in ON cells regardless of FLC levels, the mechanism behind the analog difference in FLC levels in fca-1 and fca-3 mutants remains unclear.I also understand that the authors empirically selected fca-1 and fca-3 (and fca-4 in this revision) mutants based on distinct levels of FLC (and flowering time). The main conclusion of the manuscript hinges on the different behaviors of FLC (analog) observed in these genetic backgrounds. Therefore, it is crucial to address how the various alleles of fca influence FLC regulation. However, the study did not address why different alleles of fca resulted in different analog mode of FLC expression.As a result, the study is primarily descriptive.

As explained in our answer to the Editors' Comment #1, we do not fully understand the mechanism by which the analog regulation is achieved. However, we do know that in *fca-3* a compromised/truncated version of the protein is produced, and this leads to less efficient downregulation of *FLC*. We have now added detail about the differences between the genotypes in the main text in the Introduction.

1. The study would benefit from an internal control in the FLC-venus experiments to compare fluorescence levels in ON cells with varying degrees of FLC-venus expression (after all, the authors are comparing fluorescence in different cells among different transgenic lines in different genetic backgrounds).

We have addressed this point in our response to the Editor’s Comment #2. We emphasise here again, however, that the FLC-Venus transgenic lines analysed in our study were generated by crossing, not transformation, and are therefore not independent transgenic lines.

2. I appreciate the detailed description of different alleles of fca used in this revision. However, I am still puzzled how fca-1 and fca-3, which both failed to produce "functional" FCA protein, resulted in different analog expressions of FLC. The authors discussed the analog regulation of FLC as an early regulation of FLC and the level of transcription affects the Polycomb switch. It would be more relevant to discuss how the different levels of FLC transcription are established in fca-1 and fca-3 mutants.

Due to a compromised/truncated version of the protein, *FLC* is less efficiently downregulated in *fca-3*, as discussed above. *fca-1* is equivalent to a full knockout mutant, so no FCA protein is produced and *FLC* repression does not occur, so *FLC* is well expressed. We feel that the full mechanism that initially establishes the analog regulation is beyond the scope of this study. Instead, given that analog regulation is clearly present, we focus on its interaction with the digital mode.

Reviewer #3 (Recommendations for the authors):I would like to thank first the authors for all the work performed on the previous version and the addressed comments. I find the manuscript has improved, although I still have several issues I would like to comment on:1. I appreciate the effort to study the FLC behavior in leaves. Yet, by eye it is difficult to distinguish whether Ler and fca-3 are exhibiting different behaviors, as the authors claim it is occurring in the root (Ler and fca-3 seem to express at the same levels in Figure 2S4). Hence, despite of the challenge, I believe quantification would be important to support the claims by the authors, although I am aware the authors were told by the editor this could be done 'without the need of in-depth quantification'. Otherwise, could the authors comment on that, and clarify in the manuscript?

We have now quantified FLC-Venus levels in shoot tissues for the three genotypes. The quantification reveals the same trend as we had identified by visual inspection (Figure Supp 2S4B) and which is in agreement with the data from the root tissues.

2. In my understanding, the ON transcriptional state between Ler and fca-3 is having the same (median) levels, according to Figure 2S2, which tells us that the differences between these two lines is just in the digital mode, and not the analog mode. Hence, if I understand well, digital and analog mode is not seen (or I could not clearly see) in the same line at the same time – at least, at the transcriptional level. If this is the case, I think the claim of the integration of digital and analog might need to be revised, given the current data is not showing both regulations in the same line. If this is not the case, I believe I am misunderstanding the interpretation of your data, and further clarifications would be needed.

We thank the reviewer for pointing this out. We agree that our quantification according to Figure 2S2 does not immediately reveal the transcriptional differences between Ler and *fca-3*. However, we are confident that the differences do exist and are not immediately apparent in the quantification because of a technical issue. The reason for this is that unspecific signals can be counted as real signals through automated quantification. When expression levels are high, this effect is negligible, but when expression levels are very low, the effect of unspecific signals becomes significant. As a result, the differences between Ler and *fca-3* don't appear large through our automated counting. However, we were able to verify that a real difference was present through visual inspection of the images. We conducted a blind test, where we manually counted the number of transcripts in images in which the genotype was omitted. Our, blind manual counting has revealed that no signals were counted in Ler, whereas for *fca-3* we could easily detect a small number of transcripts (2-15 mRNAs/cell) in a few cells. Thus, there is a real difference between Ler and *fca-3*, which is also evident in the FLC-Venus protein quantification analysis. For the above technical reasons, it's not as apparent through our automated image analysis pipeline.

We have explained these issues in the manuscript (in the Results and Methods sections) and we now also state that Ler quantification in smFISH is not distinguishable from the background. However, a more subtle analysis nevertheless still reveals that Ler and *fca-3* are clearly different in our smFISH results, not only in the significantly different all-cell distributions of Figure 2C, but also in the ON cell analysis of Figure 2S1/2. While ON cells in *fca-3* have similar counts as Ler "ON" cells (not real ON but false positives from noisy background), a much higher percentage of ON cells is detected in *fca-3* (Figure 2S1/2). Therefore, these ON cells detected in *fca-3* cannot be a consequence of the technical issue described above. If they were, we would expect similar frequencies of high background cells between genotypes. Hence, even from the ON cell analysis we see that *fca-3* is expressed at a higher analog level compared to Ler, and lower compared to *fca-1*.

We have also added text in the Discussion to explain how the analog/digital modes are integrated, highlighting that analog expression varies across genotypes but is predominantly stably set within each genotype over time, while digital regulation does occur over time (with the probability of digital switching affected by the analog levels). We hope this clarifies any confusion.

3. I agreed with the comment 3 by Reviewer 1, and the authors thought it is not needed to provide further evidence, given the differences between the half-lives timescales (hours) with the developmental timescales (weeks). Yet, I am wondering whether the timescale to compare to understand in part the resulting distributions should also be the division timescale (a day). Given this manuscript relies on quantitative evidences, I believe it is still unclear for the reader whether dilution effects or the protein or RNA half-lives (perhaps together with noise) might broaden the histograms, conferring the false impression of analog regulation. Could slightly different dilution (and noisy) rates between the different lines account for the different protein levels shown in Figure 2? I think given the central scope of this manuscript on distinguishing what is analog and what is digital, this point needs more justification and/or clarification (ideally, a model could help).

The correct timescale to compare the half-lives to when interpreting the gradual drop of expression and Venus intensity in the time course experiment is definitely the developmental switching timescale of 1-2 weeks. Indeed, during the process of switching there will be a short period (hours) during which the mRNA/protein levels decay in that cell, which will broaden the histograms. But at a given moment when we are measuring the mRNA/proteins levels only a small fraction of the cells (few percent) will actually be undergoing this process (~6 hr switching time out of ~10 days of growth). Hence, it cannot significantly affect histograms. We have revised the relevant section of the manuscript “FLC RNA and protein are degraded quickly relative to the cell cycle duration” to try to make this clearer. Furthermore, this effect would give the impression of analog regulation within a single genotype (hiding the digital effect) rather than affecting the comparison between genotypes where we see the analog effect.

4. I understand that the authors might find the manual fitting good enough for them, but they derive conclusions out of it, in particular now they give numbers about the OFF-ON rates in relation to the ON-OFF rates. I think that given the quantitative scope of this work, it would be more ideal to have a more reproducible fitting to have solid conclusions.

We do not agree that introducing a more complex fitting procedure will change any of our conclusions. We do give OFF-ON versus ON-OFF rates but because of the fact that cells are clearly turning off over time at a population level, there is no escaping the conclusion that the ON to OFF rate must be much higher. A more sophisticated fitting cannot change this fundamental conclusion. Furthermore, as our current fitting method provides reasonable fits to almost all our data, we do not see how making this aspect more elaborate is going to lead to any significant changes. Accordingly, and as this point was not highlighted by the editors as being an essential revision, we have therefore opted not to make any further changes on this point.

5. Overall, although I appreciate the very good and interesting work from the authors, I still find there are aspects of clarity of the manuscript that need to be revised, that better link the data from the conclusions extracted by the authors. For instance, as I said above it is not clear to me what the authors really mean by 'the integration of the digital and analog regulation', given I do not clearly see such two modes acting in the same line, and I am still hesitant about not seeing a bimodal distribution whose expression levels could be modulated, which would be the major evidence to me. Also, lines in the abstract 39-40 are not totally clear to me; I understand there is a slow time scale for the digital silencing, but I am not totally sure about associating a timescale to the analog regulation, and therefore, to compare both timescales. (Maintenance of expression levels does not necessarily show to me that analog precedes the digital silencing). I would appreciate more clarifications in this directions.

We have now revised the Discussion to better explain the integration of digital and analog. Regarding the abstract, we state that the analog precedes the digital. Indeed, we do not associate a timescale to the analog, which varies according to genotype, not time. The Discussion now clarifies this point.

6. I noticed that the manuscript version with tracked changes (which is the version I initially read in detail) is not the same as the non-tracked version (e.g. see lines 553 and 596, which refer to Sup File 3 and Sup. File 1, and compare it with the tracked version, in which the equivalent sentences would refer to Sup. File 1 and Supp Table 4, respectively). I hope there are not further major differences between the tracked and non-tracked manuscripts.

We can confirm that no further major differences exist between these files. The only changes were made to rename tables to Supp files and to remove figures from the manuscript file as we were asked to change this after the resubmission. We apologise for these differences and for the confusion this has caused.